# Mitotic chromosomes scale to nuclear-cytoplasmic ratio and cell size in *Xenopus*

Coral Y Zhou[1]*, Bastiaan Dekker[2], Ziyuan Liu[1], Hilda Cabrera[1], Joel Ryan[3],
Job Dekker[2,4], Rebecca Heald[1]*

[1]Department of Molecular and Cell Biology, University of California, Berkeley,
Berkeley, United States; [2]Department of Systems Biology, University of
Massachusetts Medical School, Worcester, United States; [3]Advanced BioImaging
Facility, McGill University, Montreal, Canada; [4]Howard Hughes Medical Institute,
Chevy Chase, United States

**\*For correspondence:**
coral.zhou@berkeley.edu (CYZ);
bheald@berkeley.edu (RH)

**Competing interest:** See page
21

**Reviewing Editor:** Jennifer
G DeLuca, Colorado State
University, United States

**Abstract** During the rapid and reductive cleavage divisions of early embryogenesis, subcellular structures such as the nucleus and mitotic spindle scale to decreasing cell size. Mitotic chromosomes also decrease in size during development, presumably to scale coordinately with mitotic spindles, but the underlying mechanisms are unclear. Here we combine in vivo and in vitro approaches using eggs and embryos from the frog *Xenopus laevis* to show that mitotic chromosome scaling is mechanistically distinct from other forms of subcellular scaling. We found that mitotic chromosomes scale continuously with cell, spindle, and nuclear size in vivo. However, unlike for spindles and nuclei, mitotic chromosome size cannot be reset by cytoplasmic factors from earlier developmental stages. In vitro, increasing nuclear-cytoplasmic (N/C) ratio is sufficient to recapitulate mitotic chromosome scaling, but not nuclear or spindle scaling, through differential loading of maternal factors during interphase. An additional pathway involving importin α scales mitotic chromosomes to cell surface area/volume ratio (SA/V) during metaphase. Finally, single-chromosome immunofluorescence and Hi-C data suggest that mitotic chromosomes shrink during embryogenesis through decreased recruitment of condensin I, resulting in major rearrangements of DNA loop architecture to accommodate the same amount of DNA on a shorter chromosome axis. Together, our findings demonstrate how mitotic chromosome size is set by spatially and temporally distinct developmental cues in the early embryo.

## Editor's evaluation

This study combines experiments in developing embryos and embryo extracts to investigate a fundamental relationship in biology – how the size of mitotic chromosomes scales with changes in cell size during development. Using the unique tools available in the *Xenopus* genus developmental biology system as well as modern genomic approaches, the authors convincingly demonstrate that mitotic chromosome scaling is mediated by differential loading of maternal chromatin remodeling factors during interphase. Although it remains unclear exactly how these factors impact chromosome size, the findings reported here will be of broad interest to the cell biology community and are likely to spawn new avenues of experimental inquiry aimed at understanding intracellular scaling relationships.

## Introduction

Upon fertilization, embryos undergo a series of rapid cell divisions in the absence of cell growth, resulting in decreasing cell size. Subcellular structures including the nucleus and mitotic spindle scale

to cell size through a defined set of mechanisms (*Heald and Gibeaux, 2018*; *Levy and Heald, 2015*). Mitotic chromosomes also shrink in size during development and scale with cell size across metazoans (*Conklin, 1912*; *Kramer et al., 2021*; *Micheli et al., 1993*), but the underlying mechanisms are poorly understood. In plants and in fly embryos, fused chromosomes with extended lengths were reported to mis-segregate during mitosis (*Schubert and Oud, 1997*; *Sullivan et al., 1993*). Similar experiments in budding yeast showed that an artificially lengthened chromosome was hyper-compacted during anaphase due to Aurora B kinase phosphorylation of substrates including condensin, a key regulator of mitotic chromosome condensation and resolution (*Neurohr et al., 2011*). In *Caenorhabditis elegans*, a screen for genes required for segregation of an extra-long, fused chromosome identified the centromeric histone CENP-A and topoisomerase II (topo II) as regulators of holocentric chromosome size (*Ladouceur et al., 2017*). However, it is unclear whether pathways that tune the length of an artificially long chromosome also operate during the physiological process of mitotic chromosome scaling during embryogenesis.

Mechanisms that scale the spindle and nucleus during development have been well-characterized. As cell volume decreases, structural components become limiting (*Good et al., 2013*; *Hazel et al., 2013*; *Levy and Heald, 2010*). Additionally, some scaling factors are regulated by the nuclear transport factor importin α, which partitions between the cytoplasm and the cell membrane and serves as a sensor for the cell's surface area to volume ratio (*Brownlee and Heald, 2019*). Previous studies of mitotic chromosome scaling, performed mainly in *C. elegans,* revealed that mitotic chromosome size correlates positively with cell size and nuclear size and negatively with intranuclear DNA density, as haploid embryos contain longer mitotic chromosomes than diploids, and knockdown of importin α or the chromatin-bound Ran guanine exchange factor RCC1 decrease both nuclear and mitotic chromosome size (*Hara et al., 2013*; *Ladouceur et al., 2015*). However, conserved relationships among genome size, nuclear size, and cell size complicate efforts to distinguish correlation from causation during mitotic chromosome scaling. Furthermore, it is unclear whether similar underlying mechanisms operate during embryogenesis of vertebrates that possess larger, monocentric chromosomes and more complex karyotypes.

The African clawed frog *Xenopus laevis* provides a powerful system for studying the mechanisms of mitotic chromosome scaling. Female frogs produce thousands of eggs that enable isolation of undiluted and cell cycle-synchronized cytoplasm in the form of egg extracts that reproduce many cellular processes in vitro including mitotic chromosome condensation and individualization (*Maresca and Heald, 2006*). In addition, fertilized eggs divide synchronously, allowing extracts to be prepared from embryos at different stages of development. Our previous work suggested that embryo nuclei added to egg extracts can recapitulate a decrease in mitotic chromosome size during development (*Kieserman and Heald, 2011*), but did not uncover underlying scaling mechanisms. It was also unclear how this observation related to scaling of mitotic chromosomes and other subcellular structures observed in vivo. Here, we fully leverage the *Xenopus* system by systematically comparing changes in mitotic chromosome size observed in vivo with perturbations in vitro to distinguish the factors that regulate mitotic chromosome scaling including nuclear size, spindle size, cell size, cell-cycle stage, and nuclear-cytoplasmic (N/C) ratio. We find that mitotic chromosomes scale continuously with spindle size, even in the largest cells of early embryos. We show that scaling occurs primarily through differential recruitment of the DNA loop extruding motor condensin I, which alters DNA loop size and thus length-wise compaction of chromosomes. Finally, we describe how reductive divisions that progressively decrease cell size and increase N/C ratio operate during different phases of the cell cycle to reduce chromosome length over the course of development. Together, these results suggest a multiscale model for how mitotic chromosome size is set in an embryo and open new avenues for deeper exploration of how changes in chromosome compaction and organization contribute to genome functions during early vertebrate embryogenesis.

## Results
### Mitotic chromosomes scale continuously with cell, nuclear and spindle size

We reasoned that mitotic chromosome size may relate to nuclear size and/or content due to chromatin factors associated with the DNA during interphase. Alternatively, mitotic chromosomes could scale with

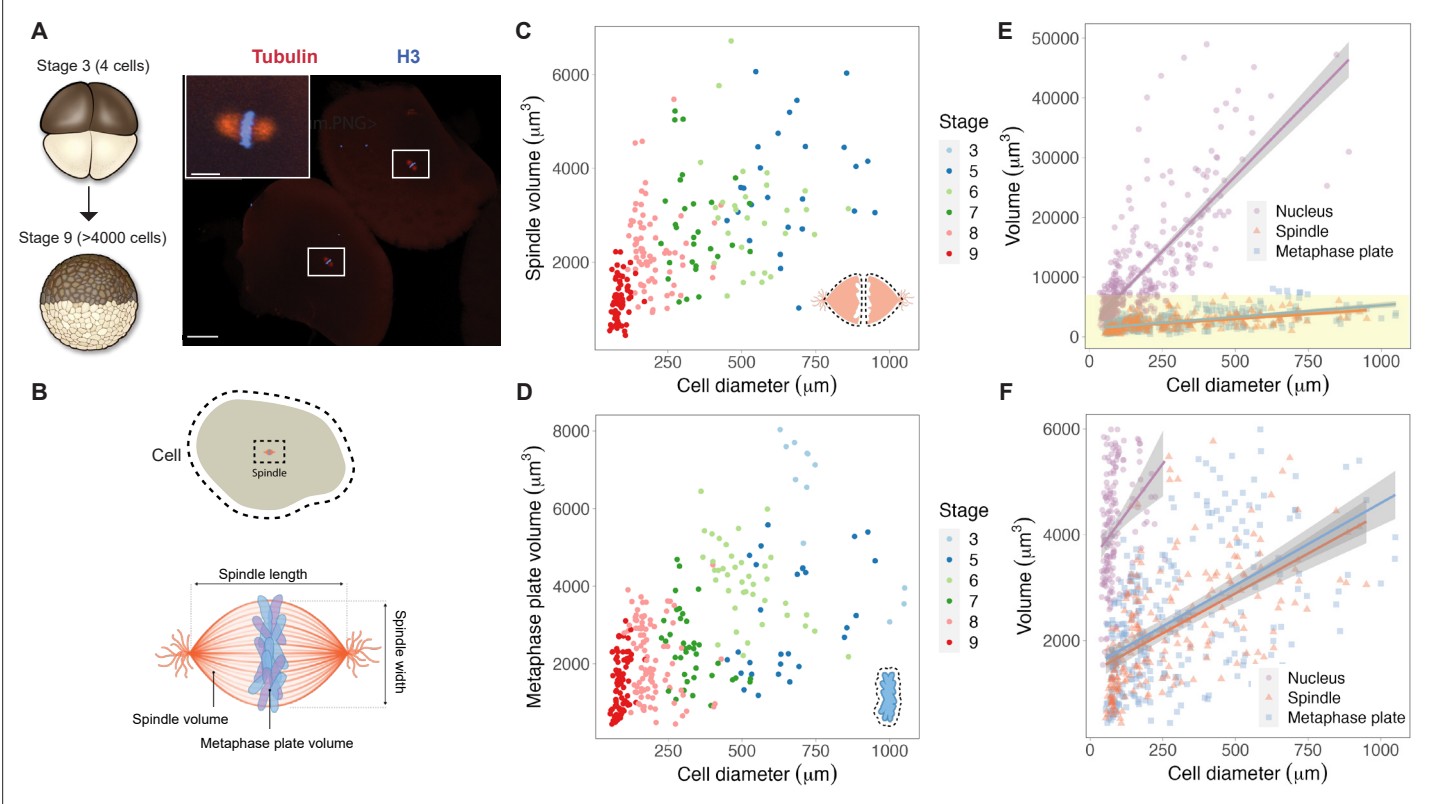

**Figure 1.** Metaphase plates scale continuously with cell size. (**A**) Experimental scheme for whole-embryo immunofluorescence. Blastula-stage embryos undergoing synchronous divisions were fixed during mitosis and stained with anti-histone H3 and anti-tubulin antibodies to visualize mitotic chromosomes and spindles, respectively. Representative image of two cells from a stage 6 embryo with white rectangles outlining mitotic spindles, scale bar = 100 μm. Inset: magnified view of one of the mitotic spindles, scale bar = 20 μm. (**B**) Dimensions of cells and spindles were either directly measured or calculated (for details, see 'Materials and methods'). (**C**) Measurements of spindle volume or (**D**) total mitotic chromosome (metaphase plate) volume plotted against cell diameter, colored by developmental stage. (**E**) Volumes of spindles, nuclei, and metaphase plates all plotted against cell diameter, fit with linear models. 95% confidence intervals shown in gray. (**F**) Zoom-in of yellow panel shown in (**E**). n = 3 biological replicates. Fold-change in size of nuclei (approximately 10-fold), spindles (approximately 2-fold), and metaphase plates (approximately 3-fold) were estimated by calculating the difference in median size in stage 3 vs. stage 8 embryos. Source data are available in *Figure 1—source data 1*.

The online version of this article includes the following source data and figure supplement(s) for figure 1:

**Source data 1.** This file contains all of the source data for Figure 1 and related supplemental figures.

**Figure supplement 1.** 3D segmentation of mitotic spindles.

**Figure supplement 2.** Scaling of spindle length vs. width to cell size.

**Figure supplement 3.** Nuclear volumes scale continuously with cell size.

**Figure supplement 4.** Metaphase plates scale similarly with both nuclei and spindles.

**Figure supplement 5.** Total chromatin volumes in interphase (nuclei) or mitosis (metaphase plates) during early cleavage divisions.

spindle size through mechanisms operating in mitosis. To distinguish between these possibilities, we performed a time course of whole-embryo immunofluorescence through the late blastula stages of *X. laevis* development and measured the dimensions of cells, spindles, and metaphase plates (*Figure 1A and B*, *Figure 1—figure supplement 1*). Although previous work showed that spindle lengths reach a plateau in cells larger than ~200 μm in diameter (*Figure 1—figure supplement 2A*; *Wühr et al., 2008*), measurement of spindle volumes by confocal microscopy revealed that spindles continue to scale at larger cell sizes (*Figure 1C*), consistent with observations that spindle width correlates more robustly with cell volume than spindle length in cultured cells (*Figure 1—figure supplement 2B*; *Kletter et al., 2021*). The combined volume of all mitotic chromosomes (the metaphase plate) also scaled continuously with cell size (*Figure 1D*), similar to published work describing nuclear scaling (*Figure 1—figure supplement 3*; *Jevtić and Levy, 2015*). To assess whether the metaphase plate

scaled more with nuclear size or with mitotic spindle size, we binned the data by cell size and plotted average volumes of the different subcellular structures. We found that total mitotic chromosome volume scaled in size remarkably similarly with both spindles and nuclei (*Figure 1—figure supplement 4*). However, the decrease in nuclear volume during early development was far greater than for mitotic structures: nuclei decreased in size approximately 10-fold over early cleavage divisions, while metaphase plate and spindle volumes decreased by approximately 3-fold and 2-fold, respectively (*Figure 1E and F*). This comparison suggests that the fold-increase in chromosome compaction as a cell transitions from interphase to metaphase diminishes from 8-fold in early blastula embryos to 1.5-fold in late blastula stages (*Figure 1—figure supplement 5*). Overall, these results demonstrate that the metaphase plate scales continuously with cell size in the early embryo, suggesting that mitotic chromosomes share scaling features with both nuclei and mitotic spindles.

## Mitotic chromosomes scale predominantly through length-wise compaction

To examine how morphologies of individual mitotic chromosomes change during development and how their size relates to metaphase plate volume, we prepared mitotic cell extracts from stage 3 (4- cell) or stage 8 (~4000- cell) embryos (*Wilbur and Heald, 2013*) and centrifuged single endogenous mitotic chromosomes onto coverslips for size measurements. We found that median chromosome lengths decreased approximately 2-fold between stage 3 and stage 8 (*Figure 2A and B*), while chromosome widths increased only by ~1.2-fold (*Figure 2—figure supplement 1*). These changes were consistent with the magnitude of metaphase plate scaling during this period estimated by whole-embryo immunofluorescence (approximately 2-3-fold, *Figure 1D*) and demonstrates that shortening of the long axis is the predominant metric underlying the change in mitotic chromosome size during early embryogenesis. We also observed that the median length of endogenous stage 3 mitotic chromosomes was not statistically different from that of replicated sperm chromosomes formed in egg extracts (*Figure 2B*), suggesting that replicated sperm chromosomes formed in egg extracts may serve as a proxy for mitotic chromosome size during the earliest cell divisions.

Previously, it was shown that mixing mitotic extracts prepared from early and late blastula stage embryos resulted in spindles of intermediate size due to equilibration of cytoplasmic spindle scaling factors (*Wilbur and Heald, 2013*). Likewise, combining interphase extracts at different ratios from two *Xenopus* species with different sized nuclei produced a graded effect on nuclear size (*Levy and Heald, 2010*). To test whether cytoplasmic factors similarly modulate mitotic chromosome size, we combined metaphase-arrested egg extracts in a 1:1 ratio with stage 8 mitotic embryo extracts containing endogenous mitotic chromosomes (*Figure 2B*). However, we observed no increase in chromosome length, indicating that mitotic chromosome scaling factors are not exchangeable in the cytoplasm during metaphase. To test whether cytoplasmic extract could alter mitotic chromosome size if added before the onset of chromosome condensation, we filtered stage 8 extracts to remove endogenous chromosomes, then added sperm nuclei either directly or following replication in interphase egg extracts (*Figure 2C*). In both cases, sperm chromosomes were at least 2-fold longer than the endogenous stage 8 chromosomes (*Figure 2D*), suggesting that stage 8 metaphase cytoplasm is sufficient to remodel sperm nuclei into mitotic chromosomes, but unable to recapitulate embryo chromosome size. Thus, mitotic chromosome size is predominantly set by factors loaded during interphase that are not readily exchangeable during metaphase, making mitotic chromosome scaling fundamentally distinct from nuclear or spindle size scaling.

## Mitotic chromosome size is determined by nuclear factors during interphase

The results above indicated that mitotic chromosome size is largely determined by nuclear rather than cytoplasmic factors. We next confirmed previous results that G2-arrested nuclei isolated from blastula-stage embryos and added to metaphase egg extracts produced mitotic chromosomes approximately 2-fold shorter than replicated sperm chromosomes formed in the same extract (*Figure 3A and B*; *Kieserman and Heald, 2011*). This finding further supports the idea that mitotic chromosome size is determined prior to entry into metaphase, likely by chromatin factors loaded during interphase. A difference in chromosome size was also recapitulated in extracts depleted of membranes through ultracentrifugation, which are incapable of forming spindles but competent for mitotic chromosome

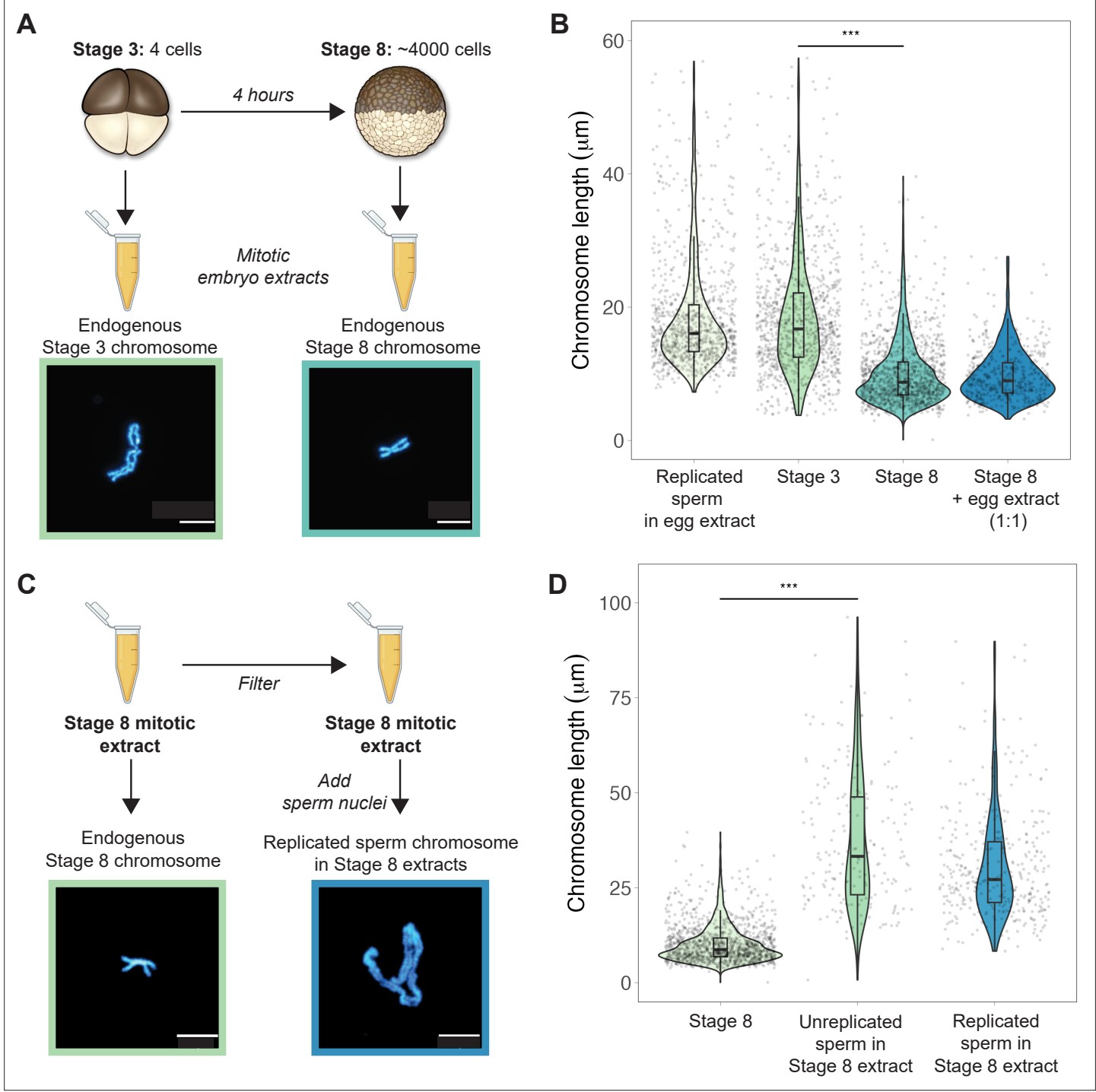

**Figure 2.** Mitotic chromosomes scale length-wise. (**A**) Mitotic extracts were prepared from stage 3 or stage 8 embryos, and single endogenous mitotic chromosomes were centrifuged onto coverslips and stained with Hoechst DNA dye. Representative images of stage 3 and stage 8 chromosomes are shown. (**B**) Length distributions of sperm mitotic chromosomes replicated in egg extract, mitotic chromosomes isolated from embryo extracts and stage 8 embryo extract chromosomes after mixing 1:1 with egg extract. (**C**) Stage 8 extracts were filtered to remove endogenous chromosomes, then unreplicated or replicated sperm nuclei were added to form mitotic chromosomes. Representative images of an endogenous stage 8 chromosome or replicated sperm chromosome formed in stage 8 extracts. (**D**) Quantification of chromosome lengths for the experiment shown in (**C**), for both replicated and unreplicated sperm conditions. n = 3 biological replicates, >50 chromosomes per replicate. Scale bar = 10 µm. ***p<0.001 by the Mann–Whitney U test. Source data are available as *Figure 2—source data 1*.

The online version of this article includes the following source data and figure supplement(s) for figure 2:

*Figure 2 continued on next page*

*Figure 2 continued*

**Source data 1.** This file contains all of the source data for Figure 2 and the related supplemental figure.

**Figure supplement 1.** Chromosome width increases slightly during early embryo cleavage stages.

assembly, indicating that spindle formation is not required for mitotic chromosome scaling (*Figure 3— figure supplement 1*). However, the magnitude decrease in chromosome size was diminished (2-fold in crude egg extracts compared to 1.4-fold in clarified extracts), suggesting that the spindle and/or membrane-associated factors may contribute to the decrease in mitotic chromosome size observed in blastula-stage embryos.

Previous work in *C. elegans* suggested that mitotic chromosome size correlates with intranuclear density and nuclear size (*Hara et al., 2013*). We observed that embryo nuclei were roughly 2-fold larger in diameter than interphase sperm nuclei (*Figure 3—figure supplement 2A*), consistent with the doubling of genome size due to the presence of both paternal and maternal genomes in embryo nuclei, suggesting that intranuclear DNA density is comparable between the two sources of nuclei. Yet, mitotic chromosomes formed by adding stage 8 embryo nuclei into egg extracts were 2-fold shorter than those formed from replicated sperm nuclei (*Figure 3B*), while mitotic spindles formed with either source of nuclei were barely distinguishable in size (*Figure 3—figure supplement 2B*). These data suggest that neither nuclear size, intranuclear DNA density during interphase, nor spindle size during metaphase determines mitotic chromosome size and further supports the idea that scaling of mitotic spindles, nuclei, and mitotic chromosomes are not necessarily coordinated.

Interestingly, although mitotic chromosome scaling could be recapitulated by adding embryo nuclei to metaphase-arrested egg extracts, chromosome morphologies were distinct. The separation of sister chromatid arms resulting in X-shaped mitotic chromosomes in both stage 3 and stage 8 mitotic embryo extracts (*Figure 2A*) was not observed when stage 8 embryo nuclei were added to egg extracts as chromosome arms remained tightly associated along their lengths (*Figure 3C*). Taken together, these results indicate that the factors determining mitotic chromosome size remain associated with chromatin in G2-arrested embryo nuclei even when introduced into metaphase egg extracts, while other factors that determine mitotic chromosome morphology do not.

## Chromosome scaling correlates with differential recruitment of condensin I, topo II, and linker histone H1.8

Robust recapitulation of chromosome scaling in metaphase-arrested egg extracts enabled molecular-level analysis of potential scaling factors, which was not technically feasible in embryo extracts that cannot transit the cell cycle in vitro. We examined three proteins known to influence chromosome size and morphology in *Xenopus*, condensin I (the predominant condensin in *Xenopus* eggs and embryos), topoisomerase II (topo II), and the maternal linker histone, termed H1.8 (*Maresca et al., 2005*; *Nielsen et al., 2020*; *Shintomi and Hirano, 2011*). After performing immunostaining of short embryo chromosomes or long sperm chromosomes formed in the same egg extracts, the abundance of each factor was calculated by normalizing immunofluorescence signal to DNA dye intensity (*Figure 3C*, see 'Materials and methods'). We found that short embryo chromosomes contained less condensin I and topo II, but more histone H1.8 relative to long replicated sperm chromosomes (*Figure 3D*, *Figure 3— figure supplement 3*). We also examined condensin II, which is 5-fold less abundant than condensin I in *Xenopus* egg extract and plays a minor role in setting mitotic chromosome size (*Ono et al., 2003*; *Shintomi and Hirano, 2011*), and similarly observed lower levels on embryo chromosomes relative to sperm chromosomes (*Figure 3—figure supplement 4*). These results are consistent with a study showing that depletion of histone H1.8 from egg extracts lengthens mitotic chromosomes (*Maresca et al., 2005*) and more recent work showing that histone H1.8 inhibits binding of condensin I and topo II to mitotic chromosomes (*Choppakatla et al., 2021*). Therefore, differential recruitment of condensins, topo II, or histone H1.8 may contribute to mitotic chromosome scaling during embryogenesis.

Our previous work showed that short embryo chromosomes could be reset to lengths observed in replicated sperm samples by cycling the mitotic chromosomes through an additional interphase in egg extracts (*Figure 3—figure supplement 5A and B*; *Kieserman and Heald, 2011*). To test whether the abundance of candidate scaling factors was affected, we performed immunofluorescence on mitotic embryo chromosomes formed during the first or second metaphase and found that the

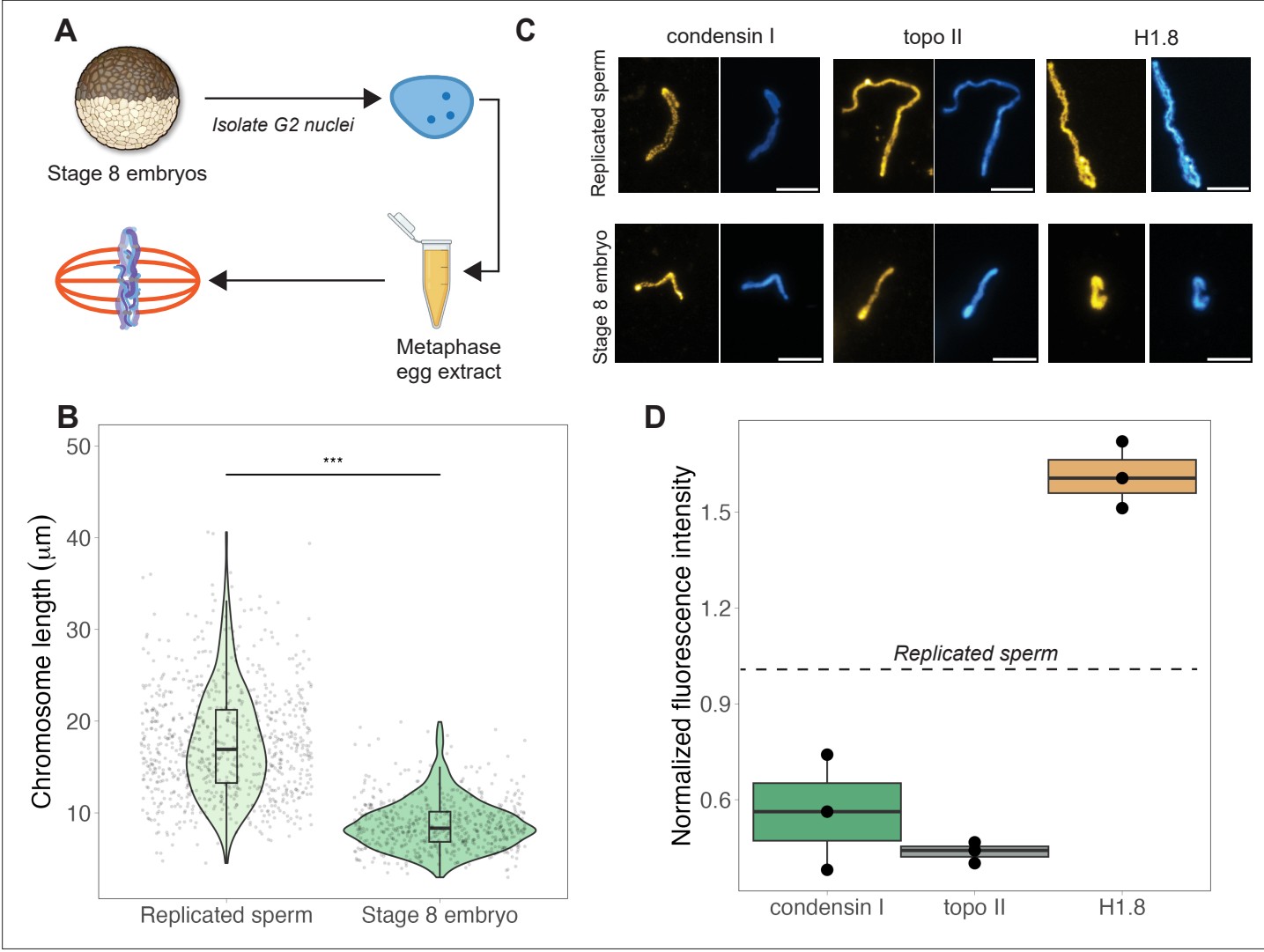

**Figure 3.** Egg extracts recapitulate mitotic chromosome scaling through differential recruitment of condensin I, topo II, and histone H1.8.
(**A**) Experimental scheme (also see 'Materials and methods'). Stage 8 embryos were arrested in G2 using cycloheximide, then fractionated to isolate cytoplasm containing nuclei. Embryo nuclei were pelleted and added to metaphase-arrested egg extracts to form mitotic spindles and chromosomes. (**B**) Lengths of replicated sperm chromosomes or stage 8 embryo chromosomes formed in metaphase egg extracts. (**C**) Representative images of mitotic chromosomes prepared by adding replicated sperm nuclei (top) or stage 8 embryo nuclei (bottom) to metaphase egg extracts, and stained with antibodies for condensin I (xCAP-G), topo II, or histone H1.8. Scale bar = 10 μm. (**D**) Abundances of topo II, condensin I, and histone H1.8 (calculated by normalizing immunofluorescence signal to Hoechst signal, see 'Materials and methods' for details) on short embryo chromosomes normalized to long sperm chromosomes (denoted by dotted line), from three different extracts. n = 3 biological replicates, >50 chromosomes per replicate. ***p<0.001 by the Mann–Whitney U test. Source data are available in *Figure 3—source data 1*.

The online version of this article includes the following source data and figure supplement(s) for figure 3:

**Source data 1.** This file contains all of the source data for Figure 3 and related supplemental figures.

**Figure supplement 1.** Clarified egg extracts support mitotic chromosome scaling.

**Figure supplement 2.** Nuclei and spindles do not scale with mitotic chromosome size in egg extracts.

**Figure supplement 3.** Raw fluorescence intensities for embryo vs. sperm immunofluorescence.

**Figure supplement 4.** Immunofluorescence of condensin II on sperm vs. embryo chromosomes formed in egg extracts.

**Figure supplement 5.** Scaling factors are reloaded onto embryo chromosomes after an additional interphase in egg extracts.

**Figure supplement 6.** Raw fluorescence intensities for anaphase experiments.

abundance of all three factors increased in the second metaphase (*Figure 3—figure supplement 5C–F*, *Figure 3—figure supplement 6*). Of the three factors, condensin I levels increased the most (2-fold), returning to levels found on replicated sperm chromosomes (*Figure 3—figure supplement 5C and D*, *Figure 3—figure supplement 6A*). Our observation that histone H1.8 levels increased slightly after the second metaphase suggests that condensin I recruitment is not necessarily regulated by H1.8, and that condensin I can override the DNA compaction activity of the linker histone to lengthen embryo chromosomes.

## Mitotic chromosomes scale through extensive remodeling of DNA loop architecture

Condensins shape mitotic chromosomes through their ability to form and extrude loops from the central axis (*Ganji et al., 2018*; *Goloborodko et al., 2016*). In silico models of loop extrusion activity suggested that tuning the abundance of condensin could dramatically alter DNA loop architecture and thus chromosome dimensions (*Goloborodko et al., 2016*). However, these models have not been tested under physiological conditions that relate to chromosome size changes in vivo.

To assess how DNA loop size and arrangement are altered in the context of mitotic chromosome scaling, we performed Hi-C on long sperm chromosomes and short embryo chromosomes formed in egg extracts. Hi-C contact maps indicated that short embryo chromosomes had increased longer-range genomic contacts along their entire length, as evidenced by thickening of the diagonal (*Figure 4A*). To quantify this effect, we plotted the decay of contact frequencies ($P$) as a function of genomic distance in base pairs ($s$) (*Figure 4B*). The shape of $P(s)$ was similar to that observed in earlier work on mitotic chromosomes from human, chicken, and *Xenopus* (*Choppakatla et al., 2021*; *Gibcus et al., 2018*; *Naumova et al., 2013*), and for rod-shaped dinoflagellate chromosomes (*Nand et al., 2021*), indicating the same overall organization of mitotic chromosomes across diverse species. Changes in the slope of $P(s)$ have been informative for modeling the underlying organization of DNA into layers of loops (*Gibcus et al., 2018*) and are more easily visualized by plotting the derivative of $P(s)$ (*Figure 4C*). Based on this previous work, the amount of DNA contained within a layer is estimated by the genomic distance at which the derivative is at its minimum, which was $10^6$ bp for sperm chromosomes compared to $\sim 10^7$ bp for embryo chromosomes (*Figure 4C*, gray bar). Within a layer, DNA loop size can be estimated from where the derivative peaks at smaller genomic distances, which was between $10^4$–$10^5$ bp for sperm chromosomes vs. $10^5$–$10^6$ bp for embryo chromosomes (*Figure 4C*, orange bar). Combined with our immunofluorescence results from *Figure 3*, these data are consistent with a model in which mitotic chromosomes scale smaller during development through decreased recruitment of condensin I, resulting in larger DNA loops and more DNA per layer, thus accommodating more DNA on a shorter chromosome axis (*Figure 4D*).

## Nuclear-cytoplasmic ratio regulates mitotic chromosome scaling, but not nuclear or spindle scaling

We next investigated possible mechanisms that could decrease the abundance of condensin I on mitotic chromosomes as they scale smaller during development. Characteristic features of cleavage divisions during early embryogenesis include a lack of cell growth and minimal gene expression, which results in exponentially increasing copies of the genome within the same total volume of cytoplasm. The increase in N/C ratio, defined here as the number of nuclei per volume of cytoplasm, titrates a finite maternal pool of DNA binding factors so that they are distributed to more and more genome copies with each subsequent cell cycle, thus lowering their abundance per genome. This effect is thought to underlie activation of zygotic transcription at the mid-blastula transition (*Amodeo et al., 2015*; *Collart et al., 2013*), and titration of the histone chaperone Npm2 was shown to play a role in nuclear scaling (*Chen et al., 2019*).

To test whether N/C ratio could play a role in mitotic chromosome scaling, we tested two different concentrations of sperm nuclei corresponding to either early (~78 sperm nuclei/µL) or late (~1250 nuclei/µL) blastula stage embryos (*Figure 5A*). After allowing the nuclei to replicate in interphase egg extracts, we added back metaphase-arrested egg extracts, isolated mitotic chromosomes for length measurements and performed immunofluorescence for condensin I, topo II, and H1.8. We found that mimicking increased N/C ratio by adding a higher concentration of sperm nuclei into egg extract decreased mitotic chromosome length ~1.3-fold (*Figure 5B*), consistent with the decrease

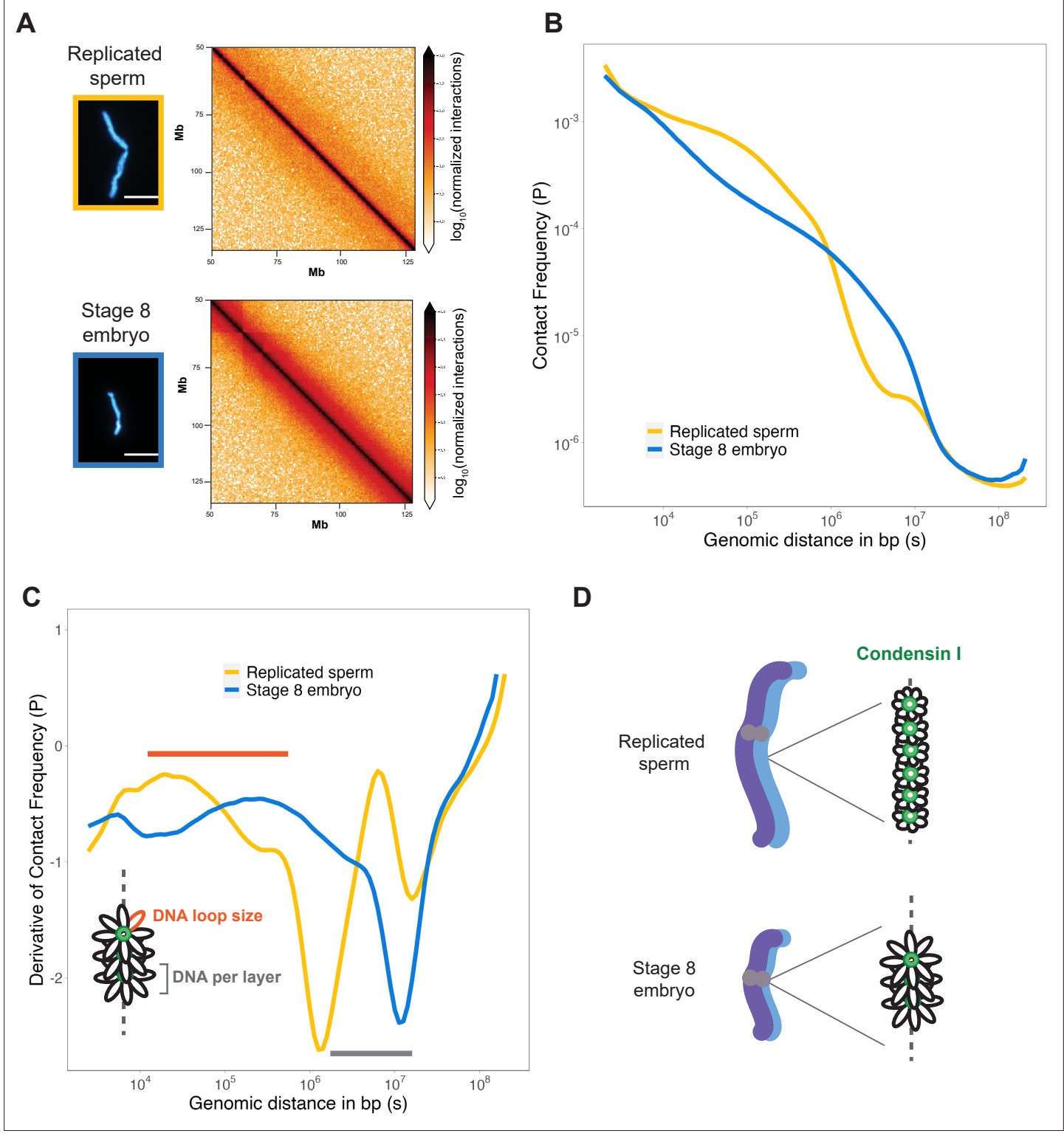

**Figure 4.** Mitotic chromosomes scale through extensive remodeling of DNA loop architecture. (**A**) Hi-C maps of chromosome 4S from replicated sperm or stage 8 embryo chromosomes formed in metaphase egg extracts. (**B**) Plots comparing how contact frequency (P) genome-wide decays as a function of genomic distance (s) for replicated sperm (yellow) or stage 8 mitotic chromosomes (blue). (**C**) Derivative of contact frequencies from (**B**). Based on previous models, peaks at $10^4$–$10^6$ bp show differences in loop size (orange bar), while inflection points at $10^6$–$10^7$ bp reveal differences in DNA amount per layer (gray bar). (**D**) Model depicting how lower condensin I occupancy on short embryo chromosomes results in an increase in DNA loop size and DNA per layer. Plots display average values from two biological replicates. Source data available as *Figure 4—source data 1*.

*Figure 4 continued on next page*

*Figure 4 continued*

The online version of this article includes the following source data for figure 4:

**Source data 1.** This file contains all of the source data for Figure 4 and related supplemental figures.

in metaphase plate size observed in vivo at the stage of development corresponding to the N/C ratios tested (stages 6–7, *Figure 1D*). This size change was accompanied by an ~1.6-fold decrease in condensin I abundance on mitotic chromosomes (*Figure 5C*), with less significant changes for histone H1.8 and topo II (*Figure 5—figure supplements 1 and 2*). Interestingly, we found that the increased N/C ratio mimicked in this experiment did not significantly affect nuclear size and increased spindle size (*Figure 5D and E*), suggesting that the N/C ratio is only capable of scaling mitotic chromosome size, but not these other subcellular structures. Previous work using lipid droplets to encapsulate spindles and nuclei in different sized compartments showed that both structures scale to compartment volume through a limiting-component mechanism (*Chen et al., 2019*; *Good et al., 2013*; *Leech et al., 2022*). However, the range of N/C ratios (defined again as genome copies per volume cytoplasm) tested here was significantly lower (approximately 10-fold) than in those studies, suggesting that maternal components do not become size-limiting for spindles and nuclei until later developmental stages.

Our previous data strongly suggested that the major molecular determinants of mitotic chromosome size are loaded during interphase. We tested this model further by varying the concentration of stage 8 embryo nuclei added to metaphase egg extracts and observed no difference in mitotic chromosome length (*Figure 5—figure supplement 3A*). This observation is consistent with the idea that titration of maternal factors had already occurred in the embryo by the time nuclei were collected at G2. Also, when sperm chromosomes were added directly to metaphase egg extracts, thus skipping the initial round of replication, there was again no effect of N/C ratio on chromosome length (*Figure 5—figure supplement 3B*). Thus, the factors that set mitotic chromosome length as a function of N/C ratio are loaded during interphase, not metaphase.

Finally, we examined whether the titration effect of chromosome factors observed in vitro corresponded to changes in their abundance during development in vivo by measuring fluorescence intensity levels of condensin I, H3, and histone H1.8 in embryo nuclei isolated at different stages. As predicted, although protein concentrations in embryos did not change over the course of the early cleavage divisions (*Figure 5—figure supplement 4A*), the levels of all factors found in interphase nuclei decreased as genome copy number increased (*Figure 5—figure supplement 4B*). Together these observations confirm that maternal factors loaded onto newly synthesized copies of the genome during interphase are titratable and demonstrate that a higher density of nuclei (increased N/C ratio) is sufficient to shorten mitotic chromosomes, likely by decreasing levels of condensin I.

## Importin α partitioning scales mitotic chromosomes to spindle and cell size

A developmental cue central to the scaling of nuclei and spindles is decreasing cell size. We previously identified importin α as a factor that coordinately scales nuclei and spindles to cell size by regulating nuclear import of lamin proteins and the activity of a microtubule destabilizing protein, respectively (*Brownlee and Heald, 2019*; *Levy and Heald, 2010*; *Wilbur and Heald, 2013*). Palmitoylation of importin α drives a fraction of the total protein to the cell membrane, where it can no longer bind to and inhibit its nuclear localization sequence (NLS)-containing cargo that regulates spindle size (*Figure 6A*). As cell size decreases during early cleavage divisions, cell surface area/volume (SA/V) increases, causing a larger fraction of importin α to associate with the plasma membrane, thus releasing more scaling factors into the cytoplasm (*Figure 6A*; *Brownlee and Heald, 2019*). To test whether importin α also plays a role in mitotic chromosome scaling, we treated egg extracts with palmostatin, an inhibitor of the major depalmitoylation enzyme APT1 to increase the pool of palmitoylated importin α, thus mimicking smaller cells with higher cell SA/V (*Figure 6B*; *Dekker et al., 2010*). We found that mitotic chromosome size decreased with palmostatin treatment and was fully rescued by the addition of purified recombinant importin α that cannot be palmitoylatated (NP importin α), but not by addition of wild-type (WT) importin α (*Figure 6B and C*). Immunofluorescence of DMSO- or palmostatin-treated chromosomes revealed increased abundance of histone H1.8 on short palmostatin-chromosomes,

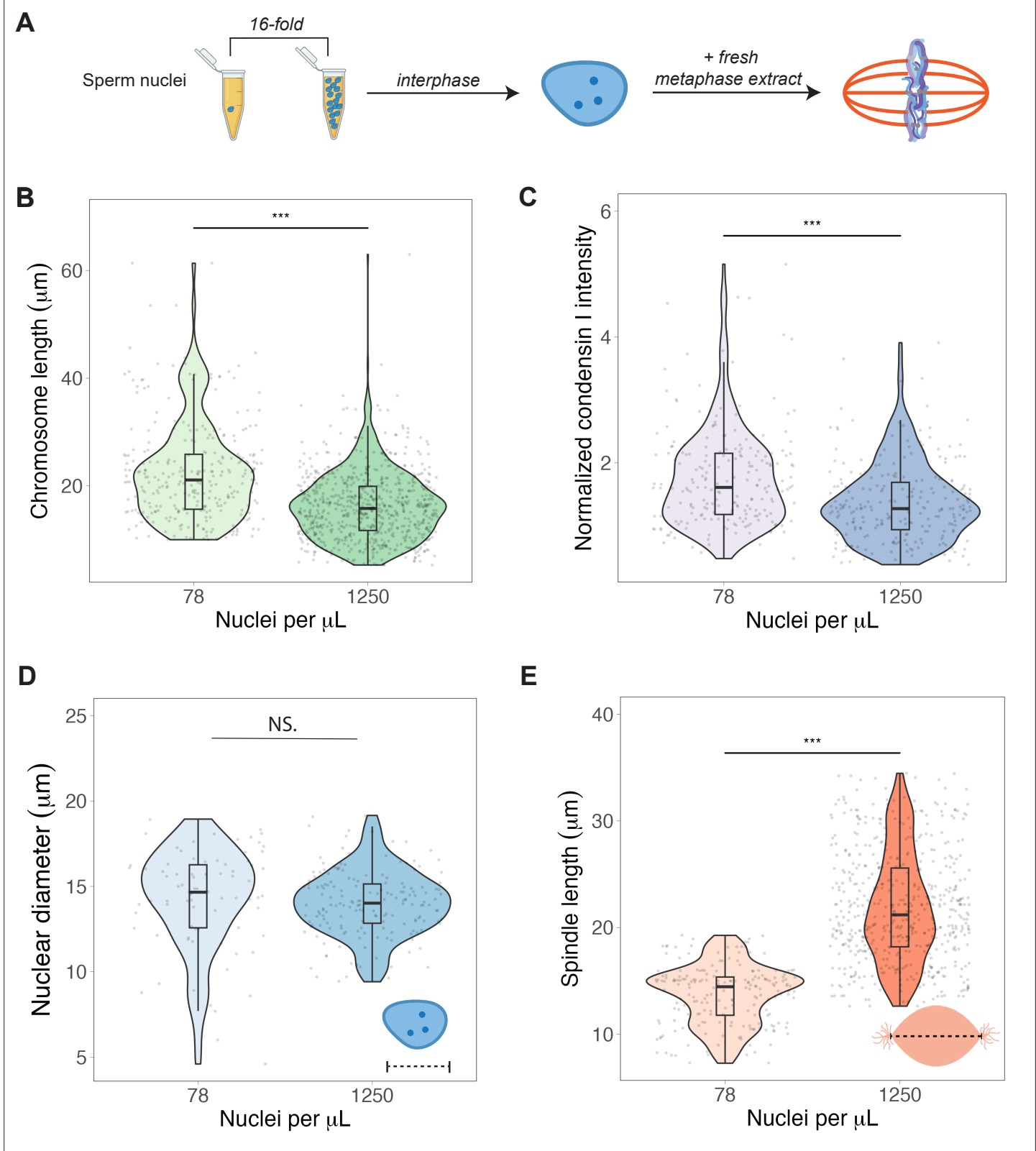

**Figure 5.** Nuclear-cytoplasmic ratio regulates mitotic chromosome scaling, but not nuclear or spindle scaling. (**A**) Concentration of sperm nuclei in egg extracts was varied 16-fold to mimic concentrations found in early (78 nuclei/μL) vs. late (1250 nuclei/μL) blastula stages. (**B**) Quantification of mitotic chromosome lengths and (**C**) condensin I abundance in extracts containing high or low concentrations of sperm nuclei (difference in median intensity is 1.6-fold). (**D**) Nuclear diameters for samples containing low or high concentrations of sperm nuclei. (**E**) Spindle lengths for samples containing low

*Figure 5 continued on next page*

*Figure 5 continued*

or high concentrations of sperm nuclei. n = 3 biological replicates, >50 structures per replicate, ***p<0.001 by the Mann–Whitney U test. NS denotes p=0.09. Source data are available in *Figure 5—source data 1*.

The online version of this article includes the following source data and figure supplement(s) for figure 5:

**Source data 1.** This file contains all of the source data for Figure 5 and related supplemental figures.

**Figure supplement 1.** Titration of topo II, histone H1.8, and H3 on mitotic chromosomes.

**Figure supplement 2.** Raw fluorescence intensities for nuclear-cytoplasmic (N/C) ratio experiments.

**Figure supplement 3.** Nuclear-cytoplasmic (N/C) ratio does not affect chromosome lengths of embryo chromosomes or unreplicated sperm chromosomes.

**Figure supplement 4.** Titration of chromatin factors during early embryogenesis.

whereas the levels of condensin I and topo II were unchanged (*Figure 6D*, *Figure 6—figure supplement 1*). These data are consistent with our observation that short embryo chromosomes contain more histone H1.8 than long sperm chromosomes (*Figure 3D*) and suggest a model in which importin α partitioning scales mitotic chromosomes to cell size by increasing availability of histone H1.8 in the cytoplasm as cell SA/V increases. Future work will be required to test whether or not H1.8 is a bona fide cargo of importin α or whether a more complicated mechanism is at play.

Our observation that condensin I levels were not altered upon palmostatin treatment suggested that scaling of mitotic chromosomes to cell size acts independently of the N/C ratio pathway. To determine when in the cell cycle importin α partitioning was affecting chromosome size, we added palmostatin to extracts either before interphase or following entry into metaphase (*Figure 6E*). Interestingly, we found that palmostatin treatment during metaphase was sufficient to shrink mitotic chromosomes, while palmostatin treatment during interphase had no effect on chromosome length (*Figure 6F*). Together, these results demonstrate that partitioning of importin α during metaphase scales mitotic chromosomes to spindle and cell size in a condensin I-independent pathway.

## Discussion

Together, our results provide new insight into how mitotic chromosome size is regulated by multiple mechanisms during the cleavage divisions of *Xenopus* embryos (*Figure 7*). Our data suggest that increasing N/C ratio during interphase is sufficient to scale mitotic chromosomes during the subsequent mitosis through decreased recruitment of condensin I, resulting in increased DNA loop and layer size and length-wise compaction. In metaphase, importin α partitioning additionally scales mitotic chromosomes to spindle and cell size due to changing cell (SA/V) ratios. Our findings have important implications for the interplay between subcellular scaling and genome structure and function during early embryogenesis.

### Mitotic chromosome size is not directly coupled to nuclear and spindle size

Previous work in *C. elegans* showed that mitotic chromosome size correlated positively with nuclear size and negatively with intranuclear density (*Hara et al., 2013*; *Ladouceur et al., 2015*). However, in egg extracts, mitotic chromosome size does not necessarily correlate with either spindle or nuclear size. G2-arrested stage 8 embryo nuclei, which contain both the maternal and paternal genomes, are almost 2-fold larger than replicated sperm nuclei (*Figure 3—figure supplement 2A*), consistent with a doubling of genome size. Yet when added to metaphase egg extracts, they form mitotic chromosomes that are 2-fold shorter than replicated sperm chromosomes (*Figure 3B*), demonstrating that in this system, mitotic chromosome size does not scale to either intranuclear density or nuclear size. Although mitotic chromosomes scaled continuously with spindle size in vivo (*Figure 1*), this correlation was abolished in vitro (*Figure 3—figure supplement 2B*). Furthermore, when N/C ratio was varied in egg extracts, mitotic chromosomes shrank 1.3-fold while spindle size increased almost 2-fold and nuclear size remained constant (*Figure 5*). Together, these data suggest that although spindles, nuclei, and mitotic chromosomes appear to scale coordinately with one another in vivo, the mechanisms regulating their size changes are distinct.

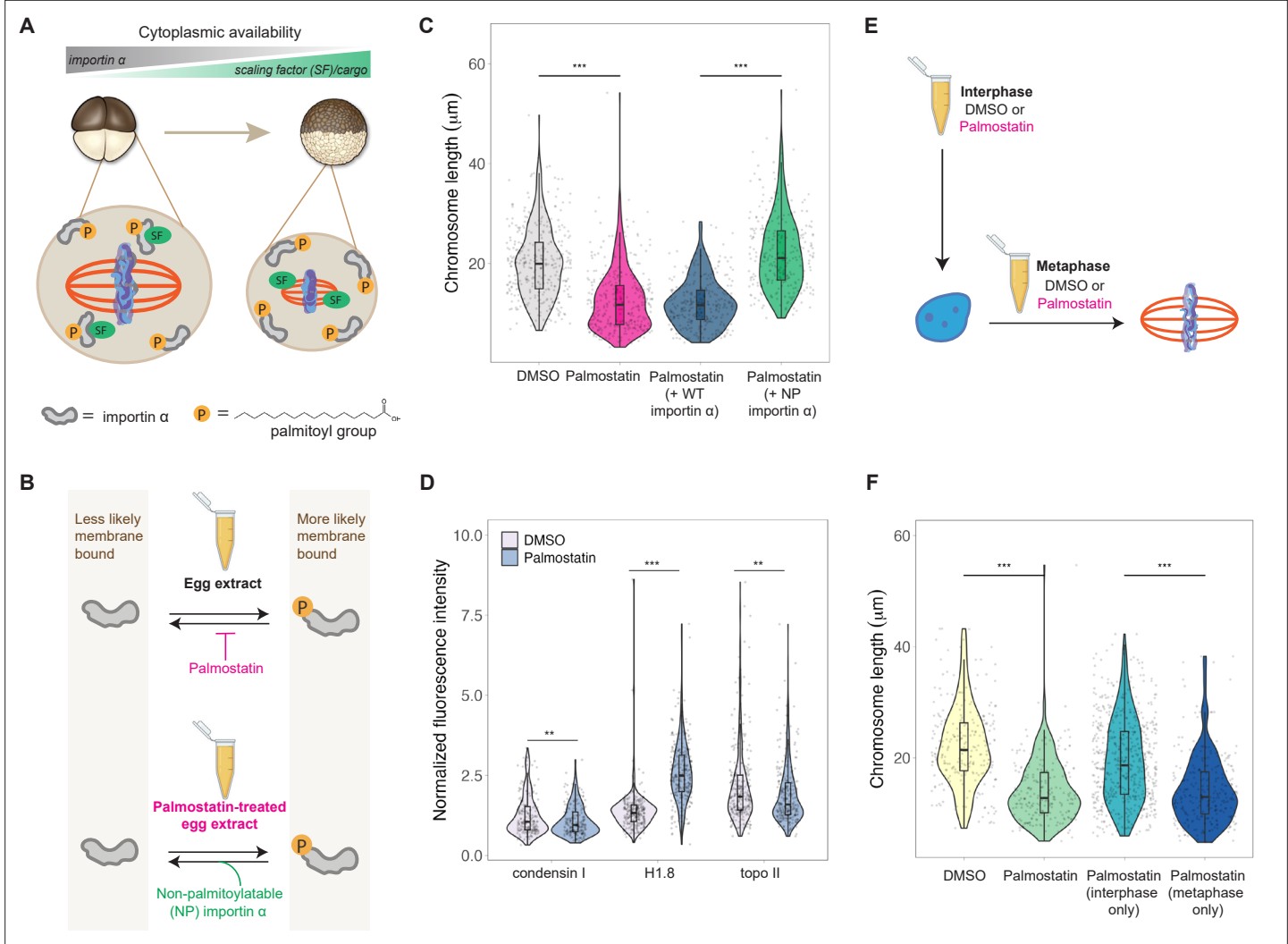

**Figure 6.** Importin α partitioning scales mitotic chromosomes to spindle and cell size. (**A**) Model for how importin α scales mitotic spindles to cell size (*Brownlee and Heald, 2019*). Due to palmitoylation of importin α, a portion of it is driven to the cell membrane, where it can no longer interact with nuclear localization sequence (NLS)-containing scaling factors, freeing them to shrink the mitotic spindle. As cell surface area/volume (SA/V) increases during embryogenesis, proportionally more importin α is driven to the membrane, thus increasing the cytoplasmic availability of scaling factors. (**B**) Top: inhibition of the major depalmitoylation enzyme APT1 by adding the drug palmostatin to egg extracts mimics smaller cells by increasing the proportion of palmitoylated, membrane-bound importin α. Bottom: addition of non-palmitoylatable (NP) importin α should rescue chromosome size in palmostatin-treated egg extracts by increasing the proportion of cytoplasmic importin α. (**C**) Quantification of mitotic chromosome lengths in palmostatin-treated extracts and rescue of chromosome length by addition of non-palmitoylatable (NP) importin α but not by wild-type (WT) importin α. (**D**) Quantification of condensin I, histone H1.8, and topo II abundance of mitotic chromosomes formed in DMSO or palmostatin-treated extracts. (**E**) Schematic of experiment to test whether importin α partitioning plays a role in chromosome scaling during interphase or metaphase. (**F**) Quantification of chromosome lengths for experiment described in (**E**). n = 3 biological replicates, >50 chromosomes per replicate. ***p<0.001 and ** <0.01 by the Mann–Whitney U test. Source data are available in *Figure 6—source data 1*.

The online version of this article includes the following source data and figure supplement(s) for figure 6:

**Source data 1.** This file contains all of the source data for Figure 6 and related supplemental figures.

**Figure supplement 1.** Raw fluorescence intensities for palmostatin experiments.

## Mitotic chromosome scaling involves temporally and spatially distinct cues

Whereas mitotic spindle size and nuclear size are set by factors operating during metaphase and interphase, respectively, we found that both phases of the cell cycle contribute to scaling of mitotic chromosomes. Experiments performed in cytoplasmic extracts showed that mitotic chromosome size is

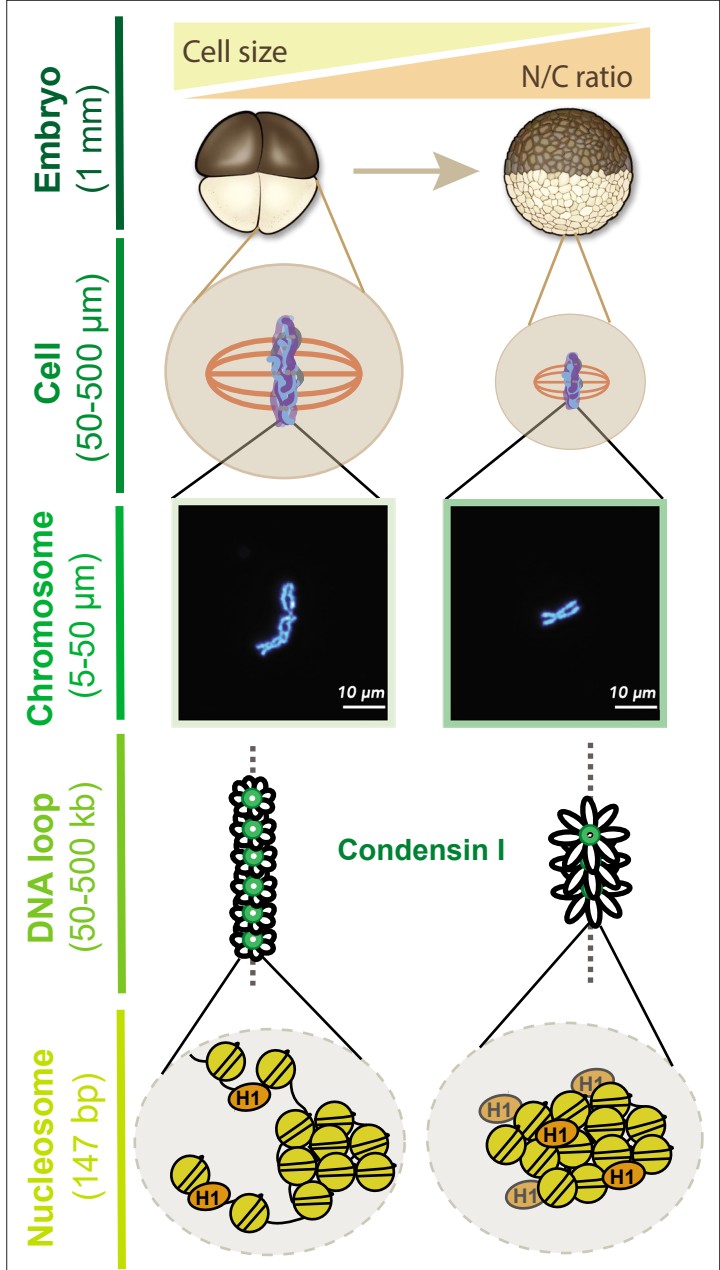

**Figure 7.** Multi-scale model for mitotic chromosome scaling. Mitotic chromosomes scale to two independent cues during development: cell size and nuclear-cytoplasmic (N/C) ratio. In the N/C ratio pathway, exponentially increasing genome copy numbers results in titration of chromatin-bound maternal factors during interphase, leading to decreased condensin I levels on metaphase chromosomes. A decrease in condensin I results in increased DNA loop size, thus allowing the same amount of DNA to be accommodated on a shorter axis. The cell size pathway additionally scales mitotic chromosomes to spindle and cell size during metaphase through increased partitioning of palmitoylated importin α to the cell membrane and release of linker histone H1.8 into the cytoplasm. We speculate that increased H1.8 could result in increased inter-nucleosomal compaction, thus creating a denser chromosome fiber.

determined by factors already present in interphase nuclei and cannot be reset by during metaphase (*Figures 2 and 3*). In contrast, importin α partitioning between the cytoplasm and plasma membrane scaled mitotic chromosomes to spindle and cell size specifically during metaphase. Consistent with this temporal separation of developmental cues, we found that condensin I acts as a scaling factor only in the N/C ratio pathway and not the importin α pathway (*Figure 5C* vs. *Figure 6D*). Together,

these data suggest that the NLS-containing cargo that scale mitotic chromosomes, unlike cargos that scale nuclei and spindles, cannot freely exchange with other factors in the egg cytoplasm to lengthen short embryo chromosomes. Since these conclusions are drawn from results obtained in vitro using reconstituted chromosomes, it remains to be determined how these two pathways would interplay in vivo. We speculate that the N/C ratio pathway that operates during interphase sets mitotic chromosome size within a certain range, but final mitotic chromosome size is set by importin α partitioning during metaphase, ensuring that metaphase plate size scales with spindle size. This idea is consistent with an early model that proposed mitotic chromosome size must be coordinated with spindle size in order to prevent chromosome mis-segregation (*Schubert and Oud, 1997*). Now that we have a more complete understanding of the pathways that regulate both spindle and mitotic chromosome size, it will be interesting to test this model in vivo. Finally, *Xenopus* embryos divide asymmetrically, with larger cells on the vegetal side. This raises the possibility that even within the same embryo the importin α and N/C ratio pathways could combine to have different effects on subcellular scaling in vivo.

## Multiple molecular pathways can regulate mitotic chromosome size

Based on in silico models of condensin I loop extrusion activity, it was predicted that changing condensin I occupancy on mitotic chromosomes would cause major changes in DNA loop size and chromosome dimensions (*Goloborodko et al., 2016*). In egg extracts, depleting condensin I by 5-fold decreased chromosome size by almost 2-fold (*Shintomi and Hirano, 2011*). However, it was unclear whether such changes occur in vivo to scale mitotic chromosomes. Our results suggest that within the physiologically relevant range of condensin I concentrations present during early embryogenesis, less condensin I correlates with both increased loop size and increased length-wise compaction. Another recent study showed that immunodepletion of H1.8 from *Xenopus* egg extracts decreased condensin I occupancy on sperm mitotic chromosomes, reducing their length (*Choppakatla et al., 2021*). In contrast, we found that an increase in condensin I could act independently of linker histone H1.8 to lengthen embryo chromosomes formed in egg extracts (*Figure 3—figure supplement 5*), suggesting that condensin I is the major scaling factor for mitotic chromosomes. The discrepancy between this study and the previous one suggests that the interplay between condensin I and H1.8 on mitotic chromosomes is more nuanced in vivo where total protein concentrations are constant during early cleavage divisions (*Figure 5—figure supplement 4*). For example, other factors could act upstream of H1.8 to regulate condensin I loading. Our importin α data suggest that chromosome size can also be modulated by H1.8 alone, without any changes in condensin I abundance (*Figure 6D*), suggesting that condensin I is not the only factor determining mitotic chromosome size in vivo. Together, our results reveal a more complex interplay among the factors that shape mitotic chromosomes than previously anticipated, some of which could also be cell-cycle dependent.

## N/C ratio as a universal mechanism for regulating chromatin structure during pre-ZGA cleavage divisions

Previous work in zebrafish and frogs suggested that progressive titration of maternal factors such as histone H3 due to increasing N/C ratio plays an important role in regulating the timing of Zygotic Genome Activation (ZGA) in the embryo (*Amodeo et al., 2015*; *Joseph et al., 2017*). Here, we recapitulated this titration effect simply by increasing the concentration of nuclei in egg extracts and found that N/C ratio is sufficient to regulate mitotic chromosome size but not spindle or nuclear size (*Figure 5*). Together, these results suggest that N/C ratio could be a universal mechanism for regulating chromatin structure across the cell cycle, but for very different functions: transcriptional regulation during interphase and chromosome segregation during metaphase. Two recent studies in *Xenopus* and zebrafish propose a model in which increasing nuclear import rates during early cleavage divisions triggers ZGA as the genome gains access to key pioneer factors that create open chromatin states permissive for transcription (*Nguyen et al., 2022*; *Shen et al., 2022*). Future studies leveraging both the in vitro and in vivo power of the *Xenopus* system will be invaluable for discovering new mechanistic links between subcellular scaling, chromatin structure, and function during early embryogenesis.

# Materials and methods

**Key resources table**

| Reagent type (species) or resource | Designation | Source or reference | Identifiers | Additional information |
|---|---|---|---|---|
| Biological sample (*Xenopus laevis*) | Male and female adult frogs | National Xenopus Resource (NXR) *Xenopus* 1 | | |
| Chemical compound, drug | Human chorionic gonadotropin hormone (hCG) | Sigma-Aldrich | CG10-10VL | |
| Chemical compound, drug | Pregnant mare serum gonadotropin (PMSG) | BioVendor | Catalog # RP1782725000 | |
| Peptide, recombinant protein | cyclinBΔ90 | *Wilbur and Heald, 2013* | | |
| Peptide, recombinant protein | UbcH10 C114S | *Wilbur and Heald, 2013* | | |
| Antibody | Anti-xCAP-G (rabbit polyclonal) | Susannah Rankin | | IF (1:500) WB (1:1000) |
| Antibody | Anti-xCAP-G2 (rabbit polyclonal) | Susannah Rankin | | IF (1:100) |
| Antibody | Anti-H1.8 (rabbit polyclonal) | *Maresca et al., 2005* | | IF (1:1000) WB (1:1000) |
| Antibody | Anti-TopoII-a (rabbit polyclonal) | Yoshiaki Azuma | | IF (1:1000) WB (1:1000) |
| Antibody | Anti-H3, Clone MABI 0301 (mouse monoclonal) | Abcam | Catalog # 39763 | Whole-embryo IF (1:500) Extract IF (1:1000) |
| Antibody | Anti-E7 beta-tubulin (mouse monoclonal) | Developmental Studies Hybridoma Bank (DSHB) | Link to product | Whole-embryo IF (1:250) |
| Antibody | Alexa Fluor 568 goat-anti-rabbit IgG (polyclonal) | Invitrogen | Catalog # A11011 | IF (1:1000) |
| Antibody | Alexa Fluor 568 goat-anti-mouse IgG (polyclonal) | Invitrogen | Catalog # A11004 | IF (1:1000) |
| Antibody | Alexa Fluor 700 goat-anti-rabbit IgG (polyclonal) | Invitrogen | Catalog # A21038 | WB (1:10,000) |
| Antibody | IRDye-800CW goat anti-mouse IgG (polyclonal) | LI-COR | Catalog # 926-32210 | WB (1:10,000) |
| Chemical compound, drug | APT1 Inhibitor (Palmostatin B) | Sigma-Aldrich | Catalog # 178501 | Reconstitute with DMSO, store at –80°C in aliquots, discard after 3 mo |
| Recombinant DNA reagent | pSNAP/Importin α (WT and NP) | *Brownlee and Heald, 2019* | | |

## In vitro fertilizations

Testes dissected from *X. laevis* males were gently cleaned with kimwipes and stored at 4°C in 1× MR (100 mM NaCl, 1.8 mM KCl, 2 mM CaCl₂, 1 mM MgCl₂, and 5 mM HEPES-NaOH pH 7.6) for 1 wk. *X. laevis* females were primed with 100 U of pregnant mare serum gonadotropin (PMSG, National Hormone and Peptide Program, Torrance, CA) at least 48 hr before use and boosted with 500 U of hCG 16 hr before an experiment. Once ovulated, females were gently squeezed along the lower spine to deposit eggs onto Petri dishes coated with 1.5% agarose in 0.1× MMR (100 mM NaCl, 2 mM KCl, 1 mM MgCl₂, 2 mM CaCl₂, 5 mM HEPES-NaOH pH 7.6, 0.1 mM EDTA). 1/4-1/3 of a *X. laevis* testes was added to 1 mL of mqH₂O in a 1.5 mL tube and homogenized using scissors and a plastic pestle for at least 30 s. The entire solution of sperm was added to a dish of eggs and swirled gently to mix. After a 10 min incubation, dishes were flooded with 0.1× MMR and incubated for an additional 10 min. Jelly coats were removed with a 3% L-cysteine solution in mqH₂O-NaOH, pH 7.8. After extensive washing

(at least five times) with 0.1× MMR, embryos were incubated at 23°C. After the first cleavage division, fertilized embryos were sorted and placed in fresh dishes containing 0.1× MMR coated with 1.5% agarose in 0.1× MMR.

## Whole-embryo immunofluorescence (Figure 1)

Successfully dividing embryos were fixed in MAD (two parts methanol, two parts acetone, one part DMSO) at 5 min intervals during each developmental stage when mitosis was likely to be occurring. After 1–3 hr of fixation, embryos were transferred to fresh MAD before storing at –20°C for up to 3 mo. Embryos were then gradually rehydrated into 0.5× SSC (75 mM NaCl, 7.5 mM sodium citrate, pH 7.0), bleached in 2% $H_2O_2$, 5% formamide and 0.5× SSC under direct light or 2–3 hr. Bleached embryos were blocked in 1× PBS, 0.1% Triton X-100, 2 mg/mL BSA, 10% goat serum, 5% DMSO for 16–24 hr at 4°C. Primary antibodies were incubated at 2–10 µg/mL at 4°C for 60 hr, washed in PBT (1× PBS, 0.1% Triton X-100, 2 mg/mL BSA) for 30 hr. Secondary antibodies were added at 2 µg/mL, covered from light and incubated for 60 hr at 4°C before washing for 30 hr with PBT. Embryos were then gradually rehydrated into 100% methanol, stored overnight at –20°C, then cleared with Murrays solution (two parts benzyl benzoate, one part benzyl alcohol). Embryos were imaged with a ×10 air objective (NA 0.45) on a Zeiss LSM 800 confocal microscope using 488 and 568 laser lines. Once cells containing a mitotic spindle were identified, z-stacks were taken at 1 µm intervals. Using Imaris, we performed 3D visualization and segmentation of mitotic spindles and metaphase plates to directly measure volumes. Cell size was measured in FIJI by manually tracing the cell in the z-stack with the largest cell area and used this measurement to calculate cell diameter. Interphase nuclear volumes were calculated from a published dataset (*Jevtić and Levy, 2015*), which used very similar methods to obtain measurements of cross-sectional areas of nuclear and cell sizes of dissociated blastomeres.

## Mitotic embryo extract preparation and sample reactions (Figure 2)

### Mitotic embryo extracts

Stage 3 and stage 8 mitotic embryo extracts were prepared as previously described (*Wilbur and Heald, 2013*). After the appropriate amount of growth at 23°C (1 hr and 45 min for stage 3 extract and 5.5 hr for stage 8 extracts), successfully dividing embryos were collected in 2 mL Eppendorf tubes, washed with five times with XB (10 mM HEPES pH 7.8, 1 mM $MgCl_2$, 0.1 mM $CaCl_2$, 100 mM KCl, 50 mM sucrose) and five times with CSF-XB (10 mM HEPES pH 7.8, 2 mM $MgCl_2$, 0.1 mM $CaCl_2$, 100 mM KCl, 50 mM sucrose, 5 mM EGTA) containing protease inhibitors LPC (10 µg/mL leupeptin, pepstatin, and chymostatin). Cytochalasin B (Cyto B) was added in the final wash for a final concentration of 20 µg/mL. Embryos were gently pelleted in a tabletop microcentrifuge at 1000 rpm for 1 min, then 2000 rpm for 30 s, at 16°C. Embryos were then crushed in a swinging bucket rotor (Sorvall HB-6) at 10,200 rpm for 12 min at 16°C. Cytoplasm was removed, placed on ice, and immediately supplemented with 10 µg/mL LPC, 20 µg/mL Cytochalasin B (cytoB), 1× energy mix (3.75 mM creatine phosphate, 0.5 mM ATP, 0.05 mM EGTA, 0.5 mM $MgCl_2$), 0.25 µM cyclinBΔ90 and 5 µM UbcH10 C114S to induce metaphase arrest, and 0.3 µM rhodamine-labeled tubulin to visualize microtubules.

### Chromosome spin-downs

All single chromosomes analyzed in this article were formed in extracts, fixed with formaldehyde, then spun onto coverslips. Previous work found that this method of fixation preserved chromosome size differences during development, unlike acetic/methanol fixation frequently used to perform karyotyping (*Kieserman and Heald, 2011*).

To examine endogenous mitotic chromosomes, 25–100 µL samples of embryo cytoplasm were incubated at 20°C for ~1 hr or until spindles had formed. Samples were then diluted fourfold in CDB (250 mM sucrose, 10 mM HEPES pH 8.0, 0.5 mM EGTA, 200 mM KCl, 1 mM $MgCl_2$) for 5–10 min, then diluted an additional fivefold in CFB containing freshly added fixative (5 mM HEPES pH 7.8, 0.1 mM EDTA, 100 mM NaCl, 2 mM KCl, 1 mM $MgCl_2$, 2 mM $CaCl_2$, 0.5% Triton X-100, 20% glycerol and 2% formaldehyde). Chromosomes were then layered on a 5 mL cushion (5 mM HEPES pH 7.8, 0.1 mM EDTA, 100 mM NaCl, 2 mM KCl, 1 mM $MgCl_2$, 2 mM $CaCl_2$ 40% glycerol) and centrifuged onto coverslips at 5500 rpm (Sorvall HS-4) for 20 min. Coverslips were removed and additionally fixed for 5 min in ice-cold methanol, and washed five times in 1× PBS, 0.1% NP-40. Coverslips were stained with Hoechst, mounted onto clean glass slides with 2 µL Vectashield, and sealed with nailpolish.

## Preparation of G2-arrested embryo nuclei (Figure 3)

### Preparation of G2-arrested embryo nuclei

Stage 8 embryos were first arrested in G2 using 150 µg/mL cycloheximide for 1.5 hr, then washed with ELB (250 mM sucrose, 50 mM KCl, 2.5 mM MgCl$_2$, 10 mM HEPES pH 7.8) containing 10 µg/mL LPC, 200 µg/mL CytoB. Embryos were gently pelleted for 1 min at 200 × $g$ in a microcentrifuge, then manually crushed with a pestle for 30 s before centrifuging at 10,000 × $g$ for 10 min. Cytoplasm was removed, placed on ice, and immediately supplemented with 10 µg/mL LPC, 20 µg/mL CytoB, 1× energy mix, and 8% glycerol. Samples were mixed gently with a cut pipet, aliquoted and flash frozen, and stored at –80°C for up to 2 y.

## Formation of mitotic chromosomes, spindles, and nuclei in egg extracts (Figure 3)

### Egg extract preparation

Egg extracts from *X. laevis* were prepared as previously described (*Maresca and Heald, 2006*). For crude extracts, eggs were packed in a clinical tabletop centrifuge and crushed in a Sorvall HB-6 rotor for 16 min at 10,200 rpm. The cytoplasm was removed and supplemented with 10 µg/mL LPC, 20 µg/mL CytoB, 1× energy mix, and 0.3 µM rhodamine-labeled tubulin. For clarified extracts, crude extracts were centrifuged at 55,000 rpm in a Ti55 rotor for 2 hr, and then 30 min to pellet membranes (all steps at 4°C). Supernatants containing soluble fraction of the cytoplasm were flash frozen and stored at –80°C for up to 3 y.

### Replicated sperm vs. embryo reactions in egg extracts

To form mitotic chromosomes from replicated sperm nuclei, purified sperm nuclei were added to 20 µL interphase egg extract at 1000 nuclei/µL unless otherwise specified. After nuclei had swelled and chromatin was replicated (around 45 min), 30 µL fresh metaphase egg extract was added and spindles formed after ~45 min. To form mitotic chromosomes from stage 8 G2-arrested embryo nuclei, nuclei were first thawed on ice for 15 min before adding 1.5 mL of CSF-XB containing 10 µg/mL LPC. Nuclei were pelleted at 1600 × $g$ for 5 min at 4°C. Nuclei pellets were resuspended in 10–15 µL of fresh CSF-XB (+LPC), added to metaphase egg extracts, and mitotic spindles formed within 1 hr. Once successful formation of mitotic spindles and chromosome condensation was confirmed by taking a small sample and staining with Hoechst dye (1 µg/mL), samples were diluted 100-fold in ice-cold 1× XBE2 (5 mM HEPES pH 7.8, 100 mM KCl, 2 mM MgCl$_2$, 0.1 mM CaCl$_2$, 5 mM EGTA, 50 mM sucrose, 0.25% Triton X-100, and 2% formaldehyde added fresh). Fixed chromosomes were layered over a 5 mL cushion containing 1× XBE2 and 30% glycerol and spun onto coverslips at 5500 rpm for 20 min at 16°C. Coverslips were removed, fixed in ice-cold methanol for 5 min, washed with 1× PBS, 0.1% NP-40 before moving on to immunostaining (see next section).

### Fixation of nuclei for size measurements

1 µL of replicated sperm nuclei or embryo nuclei was added to 3 µL of fixative (50% glycerol, 12% formaldehyde, 1× MMR, 5 µg/mL Hoechst) on a clean glass slide. A coverslip was placed on top and nuclei were imaged (see subsequent section detailing all extract imaging procedures).

### Fixation of spindles for size measurements

Spindles were fixed and spun down onto coverslips as previously described (*Maresca and Heald, 2006*). Briefly, a 50 µL extract reaction containing fully formed spindles was diluted 1:100 in fixative (80 mM Pipes, 1 mM MgCl$_2$, 1 mM EGTA, 30% glycerol, 0.5% Triton X-100, pH 6.8), followed by rocking at room temperature for up to 15 min. The fixed reaction was layered over a cushion (80 mM Pipes, 1 mM MgCl$_2$, 1 mM EGTA, 40% glycerol, 0.5% Triton X-100, pH 6.8) and centrifuged onto coverslips at 5500 rpm in a Sorvall HS-4 rotor for 20 min at 16°C. Coverslips were removed, fixed in ice-cold methanol for 5 min, washed with 1× PBS, 0.1% NP-40 before moving on to immunostaining (see next section).

## Anaphase reactions

Once mitotic spindles had formed, reactions were transferred to fresh tubes and 1× CA was added, mixed by flicking four times. After 40 min of interphase, an additional 0.5× CA was added to ensure full replication of DNA. Successful reactions were confirmed by staining a small sample with Hoechst. After 75 min in interphase, nuclei were swollen and an equal volume of fresh metaphase extract was added to induce a second round of metaphase spindle assembly. Once formed, mitotic chromosomes were fixed and isolated on coverslips using same procedures described above.

## Immunofluorescence, imaging, and analysis of chromosomes, spindles and nuclei from extracts (Figures 3, 5 and 6)

Only extracts of high quality (determined by the quality of spindles, nuclei, or chromosomes formed) were used for quantifications shown in this study.

### Immunofluorescence

Once mitotic chromosomes, nuclei, or spindles were fixed and isolated on coverslips, they were blocked overnight with 1× PBS, 3% BSA at 4°C. The rest of the procedure was performed at room temperature. Primary antibodies were added at 1–2.5 µg/mL for 1 hr and washed five times with 1× PBS, 0.1% NP-40. Secondary antibodies were added at 1 µg/mL for 1 hr, washed five times, then stained with Hoechst at 1 µg/mL for 10 min. Coverslips were washed two more times, then mounted using Vectashield without DAPI.

### Imaging

Imaging was performed on an Olympus BX51 upright epifluorescence microscope using an Olympus PlanApo ×63 oil objective for chromosomes and ×40 air objective for nuclei and spindles. Images were captured on a Hamamatsu ORCA-II camera. In order to ensure that fluorescence intensity values were comparable between samples, exposure times for each channel were set by the sample with the highest fluorescence intensity and kept constant for all replicates of that experiment.

### Selection of single chromosomes

Single chromosomes were manually selected based on having a complete morphology. Though we did not use a centromere marker, the centromere is usually evident based on punctate intense staining by condensins and topo II, and decreased staining and 'cinched' morphology by Hoechst. We also biased selection toward longer chromosomes to avoid selecting broken chromosomes. Once selected, chromosomes were cropped from the rest of the image and stored in a separate folder for further analysis.

### Analysis of chromosome dimensions and abundance of protein of interest

Chromosome lengths were measured manually using the freehand line tool. Median intensity values were used to perform background subtractions in each channel, and the abundance of a certain factor of interest was calculated by dividing the background-subtracted fluorescence intensity of the factor by the background-subtracted fluorescence intensity of Hoechst.

## Hi-C and contact probability analysis (Figure 4)

### Preparation of samples and sequencing

Hi-C was performed as previously described (*Belaghzal et al., 2017*). Mitotic chromosomes from either replicated sperm or stage 8 embryo nuclei were formed in 250 µL egg extract reactions containing 4000 nuclei/µL. Reactions were then diluted 48-fold in XBE2 containing 1% formaldehyde and 0.25% triton X-100. After 10 min of fixation with rocking at room temperature, samples were quenched for 5 min with 140 mM glycine before transferring to ice for 15 min. Chromatin was pelleted at 6000 × *g* for 20 min at 4°C, then resuspended in XBE2 containing 0.25% Triton X-100, flash frozen, and stored at –80°C. Pellets were thawed, homogenized by treatment with 0.1% SDS (final concentration), and quenched with 1% Triton X-100 (final concentration) prior to overnight digestion with 400 U DpnII at 37°C. The next day, the enzyme was inactivated prior to biotin-fill with biotin-14-dATP for 4 hr at 23°C. Subsequently, chromatin was ligated at 16°C for 4 hr. After crosslinking was reversed by proteinase K at 65°C overnight, sonicated ligation products were size selected for 100–350 bp

products. Size-selected products were end repaired followed by biotin-pull down with streptavidin. Prior to Illumina Truseq adapter ligation, purified DNA fragments were A-tailed. PCR amplification and primer removal were the last steps before final library was sequenced on Illumina HiSeq 4000 with PE50.

## Hi-C analysis

Hi-C libraries processed by mapping to the *X. laevis* 10 genome using the distiller pipeline (https://github.com/open2c/distiller-nf). Reads were aligned with bwa-mem, uniquely mapped reads were further processed after duplicate removal. Valid pair reads were binned at 1, 2, 5, 10, 25, 50, 100, 250, 500, and 1000 kb in contact matrices in the cooler format (*Abdennur and Mirny, 2019*). Cooler files were normalized using Iternative balancing correction (*Imakaev et al., 2012*), excluding first two diagonals to avoid artifacts at short range.

## Hi-C contact probability analysis

For contact probabilities, balanced Hi-C data binned at 1 kb was used to calculate contact frequency as a function of genomic distance. From cooltools (v0.5.1), expected_cis, logbin_expected, and combined_binned_expected were used to generate average contact decay plots genome-wide (*Abdennur et al., 2022*). First, the contact frequency by distance for each chromosome was calculated using expected_cis. Data was grouped into log spaced bins with logbin_expected. Genome-wide average and derivative was calculated by combined_binned_expected.

## N/C ratio experiments (Figure 5)

### Egg extract reactions

The same procedures were used as described above for egg extract experiments except anytime nuclei concentration in egg extracts was varied, the volume of extract fixed was also varied to keep the nuclei concentration per coverslip constant. We found this to be important for ensuring that any effects we observed were not due to titration of the antibody used for immunostaining.

### Western blots (*Figure 5—figure supplement 4*)

Stage 3 and stage 9 embryo extracts were prepared as described above and analyzed by Bradford to determine total protein concentrations. 25 µg protein was loaded per sample on 4–20% gradient gels (Bio-Rad). Proteins were transferred overnight at 4°C onto nitrocellulose membranes, blocked in 5% milk in Tris buffered saline containing 0.1% Triton-X100 (TBST) for 1 hr at room temperature, then stained with primary antibodies for 1 hr at room temperature. After washing with PBST 5×, blots were incubated with secondary antibodies containing infrared dyes for 1 hr at room temperature. After a final wash in PBST, blots were visualized on an LI-COR Odyssey Imager using a scanning intensities that did not overexpose the blot. Quantification of each band was performed in FIJI, and the percent abundance was calculated by normalizing to the higher intensity band for a particular frog and antibody.

### Immunofluorescence of embryo nuclei

Embryo nuclei were thawed and directly fixed in ELB supplemented with 15% glycerol and 2.6% paraformaldehyde for 15 min with rocking at room temperature. Fixed nuclei were layered over a 5 mL cushion containing 100 mM KCl, 1 mM $MgCl_2$, 100 µM $CaCl_2$, 0.2 M sucrose, and 25% glycerol. Nuclei were spun onto coverslips at 1000 × *g* for 15 min at 16°C. Coverslips were additionally fixed and washed as described above.

## Importin α experiments (Figure 6)

### Purification of importin α WT and NP

Constructs used for expression in Rosetta pLysS DE3 *Escherichia coli* were from a previous study (*Brownlee and Heald, 2019*). After reaching logarithmic growth, cells were induced with 0.4 mM IPTG overnight at 18°C. Cells were harvested and resuspended in cold lysis buffer (1× PBS, 300 mM KCl, 7.5 mM imidazole, 10% glycerol, pH 7.5) with 2 mM β-mercaptoethanol and protease inhibitors added fresh. Resuspended cells were lysed by sonication and cleared lysate was bound

to Ni-NTA resin (Thermo Scientific) for 1 hr with rocking at 4°C. Beads were batch-washed with 50 mL of wash buffer (1× PBS, 300 mM KCl, 20 mM imidazole, 10% glycerol, pH 7.5) with 2 mM β-mercaptoethanol and protease inhibitors added fresh, then packed on a 20 mL Econo-column (Bio-Rad) with 50 mL of additional wash buffer. Protein was eluted in 10 mL of 1× PBS, 100 mM KCl, 500 mM imidazole, pH 7.5, dialyzed overnight into 1× XB (100 mM KCl, 1 mM $MgCl_2$, 0.1 mM $CaCl_2$, 10 mM HEPES, 50 mM sucrose, pH 7.7) at 4° , filter concentrated, then flash frozen in liquid nitrogen, and stored at –80°C.

## Egg extract reactions

Same procedures were used as detailed above except extracts were incubated with either DMSO or 10 µM palmostatin for 45 min at 20°C before using in a reaction. During this period, reactions were thoroughly mixed by pipetting gently with a cut tip every 20 min.

## Statistical analysis and plots

All experiments shown were performed with at least three biological replicates, unless otherwise stated in figure legends. Calculations of raw data were performed in R (version 3.3.0), and final data were plotted using the *ggplot2* package (Wickham H, 2016). Negative values in fluorescence intensity measurements due to errors in imaging were removed from data before plotting. Box and violin plots depict the range of the data (first quartile, third quartile, and median), with individual datapoints in gray. All fold-differences stated in this work were calculated based on median values. As such, we used the Mann–Whitney U test to calculate significance values, which is best suited for comparisons based on medians rather than means.

## Acknowledgements

We thank Yoshiaki Azuma (University of Kansas) and Susannah Rankin (OMRF) for their generous donation of antibodies against topo II and condensins, respectively. We would like to thank J Smolka, H Cantwell, G Cavin-Meza, X Liu, and S Coyle for critical feedback on the manuscript. Some illustrations were made by Rebecca Konte and Biorender.com. Imaris software was generously donated by A Dernburg. This work was supported by NIH MIRA grant R35GM118183 and the Flora Lamson Hewlett Chair in Biochemistry to RH and a Jane Coffin Childs Memorial Fund fellowship to CYZ, an R01 from NHGRI to JD (HG003143). JD is an investigator of the Howard Hughes Medical Institute.

## Additional information

### Competing interests

Job Dekker: Reviewing editor, *eLife*. The other authors declare that no competing interests exist.

### Funding

| Funder | Grant reference number | Author |
|---|---|---|
| National Institute of General Medical Sciences | R35GM118183 | Rebecca Heald |
| Jane Coffin Childs Memorial Fund for Medical Research | | Coral Y Zhou |
| Howard Hughes Medical Institute | | Job Dekker |
| National Human Genome Research Institute | HG003143 | Job Dekker |

The funders had no role in study design, data collection and interpretation, or the decision to submit the work for publication.

## Author contributions
Coral Y Zhou, Conceptualization, Resources, Data curation, Software, Formal analysis, Supervision, Funding acquisition, Validation, Investigation, Visualization, Methodology, Writing – original draft, Project administration, Writing – review and editing; Bastiaan Dekker, Resources, Data curation, Software, Formal analysis, Investigation, Methodology, Writing – review and editing; Ziyuan Liu, Hilda Cabrera, Data curation, Formal analysis, Investigation; Joel Ryan, Software, Methodology, Writing – review and editing; Job Dekker, Resources, Supervision, Funding acquisition, Methodology, Writing – review and editing; Rebecca Heald, Conceptualization, Resources, Supervision, Funding acquisition, Methodology, Project administration, Writing – review and editing

## Author ORCIDs
Coral Y Zhou  http://orcid.org/0000-0001-7471-4645
Job Dekker  http://orcid.org/0000-0001-5631-0698

## Ethics
This study was conducted within the guidelines for Care and Use of Laboratory Animals by the National Institutes of Health. All procedures were performed in accordance with our Animal Utilization Protocol (AUP-2014-08-6596-2) and under strict regulation by the UC-Berkeley Institutional Animal Care and Use Committee (IACUC, NIH Insurance #A4107-01).

## Decision letter and Author response
Decision letter https://doi.org/10.7554/eLife.84360.sa1
Author response https://doi.org/10.7554/eLife.84360.sa2

# Additional files

## Supplementary files
• MDAR checklist

## Data availability
Sequencing data have been deposited in GEO under the code GSE217111.

The following dataset was generated:

| Author(s) | Year | Dataset title | Dataset URL | Database and Identifier |
|---|---|---|---|---|
| Zhou CY, Dekker B, Liu Z , Cabrera H, Ryan J, Dekker J, Heald R | 2022 | Mitotic chromosomes scale to nucleo-cytoplasmic ratio and cell size in *Xenopus* | http://www.ncbi.nlm.nih.gov/geo/query/acc.cgi?acc=GSE217111 | NCBI Gene Expression Omnibus, GSE217111 |

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
