## [Editor Report]

This study combines experiments in developing embryos and embryo extracts to investigate a fundamental relationship in biology – how the size of mitotic chromosomes scales with changes in cell size during development. Using the unique tools available in the *Xenopus* genus developmental biology system as well as modern genomic approaches, the authors convincingly demonstrate that mitotic chromosome scaling is mediated by differential loading of maternal chromatin remodeling factors during interphase. Although it remains unclear exactly how these factors impact chromosome size, the findings reported here will be of broad interest to the cell biology community and are likely to spawn new avenues of experimental inquiry aimed at understanding intracellular scaling relationships.

---

## [Decision Letter]

**Decision letter after peer review:**

Thank you for submitting your article "Mitotic chromosomes scale to nucleo-cytoplasmic ratio and cell size in *Xenopus*" for consideration by *eLife*. Your article has been reviewed by 3 peer reviewers, and the evaluation has been overseen by a Reviewing Editor and Jessica Tyler as the Senior Editor. The following individuals involved in the review of your submission have agreed to reveal their identity: Alexander E Kelly (Reviewer #1); Jesse C Gatlin (Reviewer #2); Paul S Maddox (Reviewer #3).

Essential revisions:

1) The authors found that in vivo chromosome volume scales with spindle size, whereas in vitro chromosome length does not. It is possible that this discrepancy is due to differences in the nature of the metrics. The chromosome volumes were calculated from the entire metaphase plate which consists of all of the chromosomes whereas the lengths were measured on individual chromosomes after dilution with buffer. Prior experiments in egg extracts (PMID: 34406118) have shown that the removal of microtubules impacts mitotic chromosome volume by inducing inter-chromosomal entanglements. These entanglements can be relieved by dilution into buffer resulting in individualized chromosomes of apparent similar length. Thus it is possible that chromosome volume is regulated in multiple ways that scale with spindle size, whereas chromosome length is set independently of spindle size. Since it is chromosome volume that changes through developmental stages in vivo, it would be important to know if the presence of the spindle can override or alter the ability of the reported factors (N/C and importin-cargo) to influence chromosome length. If chromosome volumes instead of chromosome lengths are measured in the intact spindles in the experiments performed in Figure 3 (i.e. stage 8 or replicated sperm nuclei in metaphase egg extract), are the relative differences the same between nuclei sources as seen for chromosome length? Or do the spindle forces normalize the chromosome volumes between different nuclei sources?

2) Related to the above point: There are some concerns regarding the approach used by the authors to measure spindle and metaphase plate volumes in Figure 1, which could be assuaged by providing examples of reconstructed images used to make those measurements. Concerns are based in part on previously published 3-D reconstructions of confocal images of *Xenopus* spindles taken with a higher power objective, which showed a lot of space between chromosomes in the metaphase plate. Along these same lines, it is unclear why the circumference of the cell in the plane of the spindle was used to determine cell diameter. Why not use the largest cell area found in the z-stack instead to estimate the diameter?

3) When quantifying the fluorescence intensities of factors on chromatin (eg condensin I), the authors should provide plots of the background corrected intensities separately for each channel (e.g. condensin I and DNA) in the supplementary info. It is possible that the DNA dye does not bind to the chromosomes from different sources to the same extent, and thus normalizing by DNA stain may not be the best way to cast the data.

4) In lines 444-447 the authors state that "Together these observations confirm that maternally loaded factors are titrated onto newly synthesized copies of the genome and that increasing N/C ratio is sufficient to shorten mitotic chromosomes, likely by decreasing levels of condensin I". However, in Figure 2C/D, they show that the addition of unreplicated sperm to stage 8 extract resulted in chromosome lengths that are similar to those observed when using sperm that has undergone replication in interphase extracts. This would suggest that it is something about the chromatin state (e.g. DNA methylation, histone modifications) that would need to survive maternal remodeling of the sperm chromatin rather than replication that is playing a role in setting chromosome length and condensin I levels. The authors should clarify this or provide evidence for prior replication in interphase in setting chromatid length.

5) It was sometimes hard to follow exactly how each experiment was performed due to the use of many different combinations of extracts, nuclei, and manipulations. The legends need to be more explicit or the methods section should be broken down into explicit sections for each experiment type (e.g. different nuclei in stage 8 extracts (Figure 2))

6) In regards to the two mechanisms for chromosome scaling, the logic of the abstract and flow of the paper should be made more clear, as it is essentially two stories. It would be helpful if the authors can discuss the interplay of the two mechanisms in the context of embryogenesis, as they both seem to have roughly the same effect on chromosome length in vitro (i.e. would one be predicted to have a stronger effect in vivo).

7) Based on spindle volume measurements, the authors state that size scaling occurs in cells up to 600 µm in diameter (contrary to results published by Wühr and colleagues). Little explanation is currently provided in the text, so please state how this limit was determined.

8) In Figure 1C, D, and Figure 1-S1A, B, why are the biggest cells found in stage 5 and not stage 3? This seems logically inconsistent with reductive divisions and is inconsistent with data shown from Levy and Jevtić. Please address this in the text.

9) Have karyotypes of mitotic chromosomes been published for cells at different stages of *Xenopus* development? If so, they should be referenced (and if not, a non-essential suggestion would be to include them to address single-cell variability in mitotic chromosome size).

10) The authors claim to have analyzed single chromosomes, but how was this determined in the absence of a kinetochore marker? Admittedly, in most images it's quite clear but in some, it's not (e.g. image in Figure 2C, "replicated sperm chromosome". Previous works from the Heald lab used antibodies to INCENP to mark kinetochores and identify single chromosomes unambiguously. Please clarify in the text.

11) It is unclear how similar magnitudes in fold reduction in mitotic chromosome volume from stages 3-8 and in mitotic chromosome lengths during the same stages indicate that "shortening of the long axis is the predominant metric underlying mitotic chromosome scaling during early embryogenesis". The authors should provide a more detailed rationale for this claim, particularly in the absence of any other chromosome size or shape metric.

12) The observations described in Figure 2B show that the median length of replicated sperm chromosomes formed in egg extracts was not statistically different than stage three mitotic chromosomes. The authors claim that this similarity, which might simply be a coincidence, demonstrates that replicated sperm chromosomes in egg extracts are a proxy for mitotic chromosome size during the "earliest cell divisions". It is not clear that this claim is sufficiently founded – the authors need to explain how this conclusion was reached. Also, are these observations the only basis for the statement in line 238 which reads "Robust recapitulation of chromosome scaling in metaphase arrested egg extracts enabled molecular-level analysis of potential scaling factors"? Please clarify.

13) Condensin II can bind to chromosomes throughout the cell cycle, whereas condensin I is only present on chromosomes during mitosis. While perhaps outside of the scope of the current manuscript, the authors should consider measuring the levels of condensin II on both interphase nuclei and mitotic chromosomes in the experiments in Figure 3. As condensin II has been shown to influence condensin I function and can load during interphase, this seems like a key candidate that could drive chromosome length indirectly through condensin I. Alternatively, the authors should consider discussing this possibility in the text.

14) The authors should consider a Hi-C analysis of mitotic chromosomes treated with DMSO and palmostatin. Since condensin levels remain unchanged in the experiments, some quantitative differences in the resulting Hi-C map might validate the looping model. As an alternative, the authors should consider discussing how palmostatin treatment, i.e. the importin α pathway, affects chromatin structure.

15) Others have shown using nuclei assembled in droplets of *X. laevis* egg extracts, that steady-state nuclear size is indeed dependent on the volume of extract in which the nuclei are assembled (e.g. Chen et al. 2019 & Leech et al. 2022). The authors should acknowledge these data and address the discrepancy with their results.

16) The authors do not reference Conklin 1912(or 15?) for the first reported observation of chromosome size scaling. This historical point is important and should be retained in the literature.

17) The blot in figure 5s3 is difficult to see and should be fixed.

18) Some of the graphs are pixelated and should be imported in a manner that retains readability.

19) The results in the abstract need to be more clearly stated. For instance, the second to last sentence in the abstract is vague, stating that mitotic chromosomes scale through x and y. Which way do they scale, is it larger or smaller? Maybe say scale to match cell size?

*Reviewer #1 (Recommendations for the authors):*

1) The authors found that in vivo chromosome volume scales with spindle size, whereas in vitro chromosome length does not. It is possible that this discrepancy is due to differences in the nature of the metrics. The chromosome volumes were calculated from the entire metaphase plate which consists of all of the chromosomes whereas the lengths were measured on individual chromosomes after dilution with buffer. Prior experiments in egg extracts (PMID: 34406118) have shown that the removal of microtubules impacts mitotic chromosome volume by inducing inter-chromosomal entanglements. These entanglements can be relieved by dilution into buffer resulting in individualized chromosomes of apparent similar length. Thus it is possible that chromosome volume is regulated in multiple ways that scale with spindle size, whereas chromosome length is set independently of spindle size. Since it is chromosome volume that changes through developmental stages in vivo, it would be important to know if the presence of the spindle can override or alter the ability of the reported factors (N/C and importin-cargo) to influence chromosome length. If chromosome volumes instead of chromosome lengths are measured in the intact spindles in the experiments performed in Figure 3 (i.e. stage 8 or replicated sperm nuclei in metaphase egg extract), are the relative differences the same between nuclei sources as seen for chromosome length? Or do the spindle forces normalize the chromosome volumes between different nuclei sources?

2) In lines 444-447 the authors state that "Together these observations confirm that maternally loaded factors are titrated onto newly synthesized copies of the genome and that increasing N/C ratio is sufficient to shorten mitotic chromosomes, likely by decreasing levels of condensin I". However, in Figure 2C/D, they show that the addition of unreplicated sperm to stage 8 extract resulted in chromosome lengths that are similar to those observed when using sperm that has undergone replication in interphase extracts. This would suggest that it is something about the chromatin state (e.g. DNA methylation, histone modifications) that would need to survive maternal remodeling of the sperm chromatin rather than replication that is playing a role in setting chromosome length and condensin I levels. The authors should clarify this or provide evidence for prior replication in interphase in setting chromatid length.

3) Condensin II can bind to chromosomes throughout the cell cycle, whereas condensin I is only present on chromosomes during mitosis. The authors should measure the levels of condensin II on both interphase nuclei and mitotic chromosomes in the experiments in Figure 3. As condensin II has been shown to influence condensin I function and can load during interphase, this seems like a key candidate that could drive chromosome length indirectly through condensin I.

4) When quantifying the fluorescence intensities of factors on chromatin (eg condensin I), the authors should provide plots of the background corrected intensities separately for each channel (e.g. condensin I and DNA) in the supplementary info. It is possible that the DNA dye does not bind to the chromosomes from different sources to the same extent, and thus normalizing by DNA stain may not be the best way to cast the data.

5) It was sometimes hard to follow exactly how each experiment was performed due to the use of many different combinations of extracts, nuclei, and manipulations. The legends need to be more explicit or the methods section should be broken down into explicit sections for each experiment type (e.g. different nuclei in stage 8 extracts (Figure 2)).

*Reviewer #2 (Recommendations for the authors):*

Most of my comments and concerns below address making an already solid and well-written manuscript better and simply address minor issues with the work, but all merit consideration.

Comments/Concerns/Questions for authors:

1) Based on spindle volume measurements, the authors state that size scaling occurs in cells up to 600 µm in diameter (contrary to results published by Wühr and colleagues). Little explanation is provided in the text, so I was just curious as to how this limit was determined.

2) In Figure 1C, D, and Figure 1-S1A, B, why are the biggest cells found in stage 5 and not stage 3? This seems logically inconsistent with reductive divisions and is inconsistent with data shown from Levy and Jevtić.

3) Some concerns I have about the approach used by the authors to measure spindle and metaphase plate volumes in Figure 1 would be assuaged by examples of reconstructed images used to make those measurements. My concerns are based in part on my recollection of 3-D reconstructions of confocal images of *Xenopus* spindles taken with a higher power objective, which showed a lot of space between chromosomes in the metaphase plate. Along these same lines, it is unclear to me why the circumference of the cell in the plane of the spindle was used to determine cell diameter. Why not use the largest cell area found in the z-stack instead to estimate the diameter?

4) Have karyotypes of mitotic chromosomes been published for cells at different stages of *Xenopus* development? If so, they should be referenced, if not, is it worth including them? I think that there might be some value in this, as I am curious about single-cell variability in mitotic chromosome size.

5) The authors claim to have analyzed single chromosomes, but how was this determined in the absence of a kinetochore marker? Admittedly, in most images it's quite clear but in some, it's not (e.g. image in Figure 2C, "replicated sperm chromosome". Previous works from the Heald lab used antibodies to INCENP to mark kinetochores and identify single chromosomes unambiguously.

6) It is unclear to me how similar magnitudes in fold reduction in mitotic chromosome volume from stages 3-8 and in mitotic chromosome lengths during the same stages indicate that "shortening of the long axis is the predominant metric underlying mitotic chromosome scaling during early embryogenesis". The authors should provide a more detailed rationale for this claim, particularly in the absence of any other chromosome size or shape metric.

7) The observations described in Figure 2B show that the median length of replicated sperm chromosomes formed in egg extracts was not statistically different than stage three mitotic chromosomes. The authors claim that this similarity, which might simply be a coincidence, demonstrates that replicated sperm chromosomes in egg extracts are a proxy for mitotic chromosome size during the "earliest cell divisions". I'm not sure that this claim is sufficiently founded. Also, are these observations the only basis for the statement in line 238 which reads "Robust recapitulation of chromosome scaling in metaphase arrested egg extracts enabled molecular-level analysis of potential scaling factors"?

8) Is there any way to use ultrastructural analysis to confirm the DNA looping model, that is that less condensin results in larger loop size and thus shorter chromosomes?

9) The authors should consider a Hi-C analysis of mitotic chromosomes treated with DMSO and palmostatin. Since condensin levels remain unchanged in the experiments, some quantitative differences in the resulting Hi-C map might validate the looping model. In the absence of this, they might want to consider discussing how palmostatin treatment, i.e. the importin α pathway, affects chromatin structure.

10) Others have shown using nuclei assembled in droplets of *X. laevis* egg extracts, that steady-state nuclear size is indeed dependent on the volume of extract in which the nuclei are assembled (e.g. Chen et al. 2019 & Leech et al. 2022). The authors should acknowledge this data and address the discrepancy with their results.

*Reviewer #3 (Recommendations for the authors):*

In the paper by Zhou, et.al., the authors set forth to build a molecular understanding of chromosome size scaling in the non-holocentric *Xenopus* system. Taking advantage of the unique toolset in this system, the authors conclude nicely that condensin I levels directly correlate with chromosome size. Further, importin levels decrease leading to axial shortening of chromosomes during development. The combined physical outcome is that Mitotic Chromatin looping changes, resulting in axial compression of chromosomes. I have no major issues with this paper.

---

## [Author Response]

Essential revisions:1) The authors found that in vivo chromosome volume scales with spindle size, whereas in vitro chromosome length does not. It is possible that this discrepancy is due to differences in the nature of the metrics. The chromosome volumes were calculated from the entire metaphase plate which consists of all of the chromosomes whereas the lengths were measured on individual chromosomes after dilution with buffer. Prior experiments in egg extracts (PMID: 34406118) have shown that the removal of microtubules impacts mitotic chromosome volume by inducing inter-chromosomal entanglements. These entanglements can be relieved by dilution into buffer resulting in individualized chromosomes of apparent similar length. Thus it is possible that chromosome volume is regulated in multiple ways that scale with spindle size, whereas chromosome length is set independently of spindle size. Since it is chromosome volume that changes through developmental stages in vivo, it would be important to know if the presence of the spindle can override or alter the ability of the reported factors (N/C and importin-cargo) to influence chromosome length. If chromosome volumes instead of chromosome lengths are measured in the intact spindles in the experiments performed in Figure 3 (i.e. stage 8 or replicated sperm nuclei in metaphase egg extract), are the relative differences the same between nuclei sources as seen for chromosome length? Or do the spindle forces normalize the chromosome volumes between different nuclei sources?

We thank the reviewers for raising this important point. We have now measured metaphase plate volumes in spindles formed from either replicated sperm or stage 8 embryo nuclei in egg extracts. Since embryo nuclei contain both the maternal and paternal copies of the genome, thus doubling chromosome number compared to replicated sperm nuclei, we reasoned that if the 2-fold decrease in chromosome length observed at stage 8 was retained in the context of metaphase plate volumes in an intact spindle formed in vitro, we would expect to observe similar metaphase plate volumes in the two samples. However, we measured a ~1.3-fold increase (over 4 biological replicates) in metaphase plate volume in spindles formed using embryo nuclei compared to those formed with sperm nuclei, indicating that total embryo chromosome volumes are greater than expected in vitro compared to in vivo (see Author response image 1).

These results suggest that the spindle may indeed influence mitotic chromosome size in egg extracts, but there are several technical problems that complicate interpretation of this experiment. Unlike in vivo structure sizes quantified in Figure 1, chromosomes in spindles formed in egg extracts are more heterogenous, often misaligning at the metaphase plate, especially the long, replicated sperm chromosomes. Thus, we cannot rule out the possibility that the difference in metaphase plate volumes we observe is actually due to differences in chromosome alignment and/or inter-chromosomal packing, both which would affect segmentation of metaphase plates. Furthermore, the buffers and procedures used to fix and spin down single chromosomes vs. whole spindles are different, which could also complicate interpretation of this new data in the context of single-chromosome measurements. Due to these caveats, which would be laborious and distracting to explain in the main text, we have decided not to include these new data in the paper. Instead, we have clearly distinguished experiments comparing metaphase plate volumes from single chromosome measurements and do not draw any quantitative conclusions when comparing in vivo and in vitro data (also mentioned in reviewer point 2).

**Author response image 1. sa2fig1:** Metaphase plate volume measurements of stage 8 embryo nuclei and sperm nuclei within spindles formed in egg extract. Since embryo nuclei contain both maternal and paternal genomes, a similar metaphase plate size would indicate a 2-fold decrease in volume for embryo chromosomes. However, median embryo metaphase plate volumes are 1.3fold larger, indicating that metaphase plate volumes follow the same trend but do not quantitatively recapitulate scaling of individual mitotic chromosomes. See text above for additional explanation. n = 4 biological replicates.

2) Related to the above point: There are some concerns regarding the approach used by the authors to measure spindle and metaphase plate volumes in Figure 1, which could be assuaged by providing examples of reconstructed images used to make those measurements. Concerns are based in part on previously published 3-D reconstructions of confocal images of *Xenopus* spindles taken with a higher power objective, which showed a lot of space between chromosomes in the metaphase plate. Along these same lines, it is unclear why the circumference of the cell in the plane of the spindle was used to determine cell diameter. Why not use the largest cell area found in the z-stack instead to estimate the diameter?

We agree with the reviewers that using metaphase plate volume measurements as a proxy for degree of mitotic chromosome condensation is potentially complicated by inter-chromosomal space. The imaging in this experiment was under low magnification (10x objective), so we did not observe any obvious holes in the metaphase plate, but it is likely that they do exist. This issue emphasizes the value of our approach of using egg and embryo extracts to examine how individual mitotic chromosomes scale, rather than drawing conclusions based on in vivo metaphase plate measurements. We have now provided example z-stacks and a 3D reconstruction of our segmentation in the supplement (Figure 1—figure supplement 1). We have also removed all language that quantitatively compares the results of the whole embryo IF experiments in Figure 1 with the scaling of individual mitotic chromosomes in vitro described in subsequent figures.

The previous method description was incorrect. We indeed used the z-stack with the largest cell area to calculate cell diameter which is now corrected in the methods section.

3) When quantifying the fluorescence intensities of factors on chromatin (eg condensin I), the authors should provide plots of the background corrected intensities separately for each channel (e.g. condensin I and DNA) in the supplementary info. It is possible that the DNA dye does not bind to the chromosomes from different sources to the same extent, and thus normalizing by DNA stain may not be the best way to cast the data.

We have now added background corrected raw intensities for each channel as supplemental figures (Figure 3—figure supplement 3, 4, 6; Figure 5—figure supplement 2, Figure 6—figure supplement 1).

4) In lines 444-447 the authors state that "Together these observations confirm that maternally loaded factors are titrated onto newly synthesized copies of the genome and that increasing N/C ratio is sufficient to shorten mitotic chromosomes, likely by decreasing levels of condensin I". However, in Figure 2C/D, they show that the addition of unreplicated sperm to stage 8 extract resulted in chromosome lengths that are similar to those observed when using sperm that has undergone replication in interphase extracts. This would suggest that it is something about the chromatin state (e.g. DNA methylation, histone modifications) that would need to survive maternal remodeling of the sperm chromatin rather than replication that is playing a role in setting chromosome length and condensin I levels. The authors should clarify this or provide evidence for prior replication in interphase in setting chromatid length.

We agree with the reviewer’s interpretation of our results presented here, except that we do not specifically invoke replication as the process during interphase that resets chromosome length. To clarify this point, we have added new data showing that adding sperm nuclei at two different N/C ratios directly to metaphase egg extracts, thus skipping interphase, does not lead to any change in chromosome size (Figure 5—figure supplement 4B). These data suggest that replication is an important step in setting mitotic chromosome size in the ensuing metaphase, but more work will be required to confirm this hypothesis, which is beyond the scope of this manuscript.

5) It was sometimes hard to follow exactly how each experiment was performed due to the use of many different combinations of extracts, nuclei, and manipulations. The legends need to be more explicit or the methods section should be broken down into explicit sections for each experiment type (e.g. different nuclei in stage 8 extracts (Figure 2))

To address this point, we have edited the text for clarity and re-organized the methods section to match the order of experiments presented in the paper, with the figure number explicitly labeled in the title of each section.

6) In regards to the two mechanisms for chromosome scaling, the logic of the abstract and flow of the paper should be made more clear, as it is essentially two stories. It would be helpful if the authors can discuss the interplay of the two mechanisms in the context of embryogenesis, as they both seem to have roughly the same effect on chromosome length in vitro (i.e. would one be predicted to have a stronger effect in vivo).

We agree with the reviewers that examining the interplay between the two mechanisms, ideally in an embryo, is a worthwhile and fascinating future direction. Since the manipulations that led to these two models were all performed in vitro, we hesitate to interpret the magnitude of the two effects as potentially additive in vivo, and rather simply state that the N/C ratio pathway is set during interphase and the cell size pathway is set during metaphase. We added some speculation that perhaps mitotic chromosome size is primarily set by the N/C ratio pathway, but the importin a partitioning pathway ensures that mitotic chromosome size is coordinated with spindle size, which has been proposed to be important to prevent chromosome missegregation. We have edited the discussion to include some of these ideas (page 16, lines 387392).

7) Based on spindle volume measurements, the authors state that size scaling occurs in cells up to 600 µm in diameter (contrary to results published by Wühr and colleagues). Little explanation is currently provided in the text, so please state how this limit was determined.

We agree with the reviewers that the 600 µm limit was somewhat arbitrary. We have now removed this number from the main text (line 106) and modified our language to focus on our major conclusion that scaling of spindles in 3 dimensions does not appear to reach an upper limit as previously reported in *Xenopus* for spindle length (Wühr et al., 2008., Figure 1— Supplement 2A).

8) In Figure 1C, D, and Figure 1-S1A, B, why are the biggest cells found in stage 5 and not stage 3? This seems logically inconsistent with reductive divisions and is inconsistent with data shown from Levy and Jevtić. Please address this in the text.

We agree with the reviewers that this is counterintuitive. Measuring cell size in stage 5 and stage 3 embryos is significantly more challenging than in later stages due to the thickness of the sample and the elongated shape of the cells, making it more difficult to approximate them as spheres. Jevtić and Levy had performed these measurements on dissociated blastomeres rather than in intact embryos, which helped alleviate these challenges.

To address this point, we have now included a third biological replicate in our data and reanalyzed all of our stage 3 and 5 samples. Taken together, these new data show that despite some variability, the largest cells are found in stage 3 embryos.

9) Have karyotypes of mitotic chromosomes been published for cells at different stages of *Xenopus* development? If so, they should be referenced (and if not, a non-essential suggestion would be to include them to address single-cell variability in mitotic chromosome size).

A previous study from our lab (Kieserman and Heald, 2011) found that using established methods to prepare karyotypes from cleavage stage embryos, which requires generating chromosome spreads though harsh fixation with acetic acid and methanol, distorted mitotic chromosome morphology such that scaling was not observable in the early embryo. The only published karyotypes we found compared blastula to the tadpole stage, which is days later in development (Micheli et al., 1993). Thus, in vitro systems using egg and embryo extracts amenable to more gentle fixatives is our method of choice for observing physiological single chromosome scaling during cleavage divisions that recapitulates the overall magnitude of metaphase plate scaling in vivo. We have stated this logic explicitly in the methods section (Page 22, lines 494-497).

10) The authors claim to have analyzed single chromosomes, but how was this determined in the absence of a kinetochore marker? Admittedly, in most images it's quite clear but in some, it's not (e.g. image in Figure 2C, "replicated sperm chromosome". Previous works from the Heald lab used antibodies to INCENP to mark kinetochores and identify single chromosomes unambiguously. Please clarify in the text.

We have now added a section in the methods that explicitly states how single chromosomes were chosen for analysis (Page 26, lines 591-594).

11) It is unclear how similar magnitudes in fold reduction in mitotic chromosome volume from stages 3-8 and in mitotic chromosome lengths during the same stages indicate that "shortening of the long axis is the predominant metric underlying mitotic chromosome scaling during early embryogenesis". The authors should provide a more detailed rationale for this claim, particularly in the absence of any other chromosome size or shape metric.

We now include measurements of chromosome widths of stage 3 and stage 8 chromosomes isolated from embryo extracts (Figure 2—figure supplement 1) and show that chromosomes increase slightly (1.2-fold) in width as they scale smaller. We have now modified the text to include these data, which support our conclusion that chromosomes scale smaller during development primarily through length-wise compaction.

12) The observations described in Figure 2B show that the median length of replicated sperm chromosomes formed in egg extracts was not statistically different than stage three mitotic chromosomes. The authors claim that this similarity, which might simply be a coincidence, demonstrates that replicated sperm chromosomes in egg extracts are a proxy for mitotic chromosome size during the "earliest cell divisions". It is not clear that this claim is sufficiently founded – the authors need to explain how this conclusion was reached. Also, are these observations the only basis for the statement in line 238 which reads "Robust recapitulation of chromosome scaling in metaphase arrested egg extracts enabled molecular-level analysis of potential scaling factors"? Please clarify.

We thank the reviewers for reminding us of this important point. To reiterate our logic, in vivo measurements demonstrated scaling of metaphase plate volumes, with a 2-3-fold decrease during early cleavage divisions (Figure 1), and consistent with observations that single endogenous mitotic chromosomes isolated from stage 8 embryo extracts were 2-fold shorter than those isolated from stage 3 embryos (Figure 2). Both the fold difference and the distribution of absolute lengths were preserved when substituting sperm chromosomes for stage 3 embryo nuclei in vitro. We agree that comparing stage 3 and stage 8 embryo nuclei would be preferable, but because of the low number of nuclei (4!) per embryo at this stage, we could not easily compare endogenous mitotic chromosomes by imaging, since it was impossible equalize nuclei concentration between stage 3 and stage 8 samples on coverslips, which we empirically discovered to be important for preventing artefacts during immunofluorescence. Nor could we isolate enough nuclei from stage 3 for in vitro reactions. Thus, reviewers are correct that we cannot rule out inherent differences between replicated sperm and embryo chromatin.

For these reasons, performing immunofluorescence of sperm vs. embryo chromosomes formed in egg extracts was the best that we could do considering technical limitations. We have softened the language to state that sperm chromosomes may serve as a proxy for early embryo chromosomes (page 6, lines 136-137) and now mention this caveat to our methods in the discussion (page 16, lines 385-387).

13) Condensin II can bind to chromosomes throughout the cell cycle, whereas condensin I is only present on chromosomes during mitosis. While perhaps outside of the scope of the current manuscript, the authors should consider measuring the levels of condensin II on both interphase nuclei and mitotic chromosomes in the experiments in Figure 3. As condensin II has been shown to influence condensin I function and can load during interphase, this seems like a key candidate that could drive chromosome length indirectly through condensin I. Alternatively, the authors should consider discussing this possibility in the text.

We thank the reviewers for this suggestion and now include data showing that embryo chromosomes recruit less condensin II than replicated sperm chromosomes in egg extracts (Figure 3—figure supplement 4).

14) The authors should consider a Hi-C analysis of mitotic chromosomes treated with DMSO and palmostatin. Since condensin levels remain unchanged in the experiments, some quantitative differences in the resulting Hi-C map might validate the looping model. As an alternative, the authors should consider discussing how palmostatin treatment, i.e. the importin α pathway, affects chromatin structure.

We have now performed immunofluorescence for additional scaling factors on DMSO- and palmostatin-treated chromosomes, revealing that the only factor that changed significantly was histone H1.8 (Figure 6D). Based on our model of how importin a partitioning regulates subcellular scaling, this result suggested that H1.8 could be a potential mitotic cargo. In this model, as cell SA/V increases during cleavage divisions, a greater fraction of palmitoylated importin a associates with the cell membrane, freeing additional H1.8 to shrink mitotic chromosomes. Our finding that short embryo chromosomes have higher levels of H1.8 than long sperm chromosomes is also consistent with this model. However, it remains unclear whether H1.8 is a bona fide importin a cargo in egg extracts, since importin 7/importin β has been reported to transport linker histones to the nucleus (https://pubmed.ncbi.nlm.nih.gov/10228156/). Further work will be required to confirm if H1.8 is a bona fide cargo of importin a, or if a more complicated mechanism is at play.

15) Others have shown using nuclei assembled in droplets of *X. laevis* egg extracts, that steady-state nuclear size is indeed dependent on the volume of extract in which the nuclei are assembled (e.g. Chen et al. 2019 & Leech et al. 2022). The authors should acknowledge these data and address the discrepancy with their results.

We now include these references when interpreting our nuclear titration results (page 12 lines 284-289).

16) The authors do not reference Conklin 1912(or 15?) for the first reported observation of chromosome size scaling. This historical point is important and should be retained in the literature.

We now include this reference in the introduction (line 46).

17) The blot in figure 5s3 is difficult to see and should be fixed.

We have repeated the Western blots with two biological replicates and now include clearer images in the manuscript (Figure 5—figure supplement 4A).

18) Some of the graphs are pixelated and should be imported in a manner that retains readability.

We have re-loaded every plot as high-resolution PNG files in this draft.

19) The results in the abstract need to be more clearly stated. For instance, the second to last sentence in the abstract is vague, stating that mitotic chromosomes scale through x and y. Which way do they scale, is it larger or smaller? Maybe say scale to match cell size?

We have altered the language in the abstract and throughout the paper to be more precise when referring to scaling.